# How Carvedilol activates β₂-adrenoceptors

Tobias Benkel [1,2], Mirjam Zimmermann[3], Julian Zeiner[1], Sergi Bravo [1], Nicole Merten [1], Victor Jun Yu Lim[4], Edda Sofie Fabienne Matthees [5], Julia Drube [5], Elke Miess-Tanneberg[6], Daniela Malan[7], Martyna Szpakowska [8], Stefania Monteleone[4], Jak Grimes[9], Zsombor Koszegi [9], Yann Lanoiselée[9], Shannon O'Brien [9], Nikoleta Pavlaki[10], Nadine Dobberstein[3], Asuka Inoue [11], Viacheslav Nikolaev[10], Davide Calebiro [9], Andy Chevigné [8], Philipp Sasse [7], Stefan Schulz[6,12], Carsten Hoffmann[5], Peter Kolb [4], Maria Waldhoer[3,13], Katharina Simon[1], Jesus Gomeza[1] & Evi Kostenis [1] ✉

Carvedilol is among the most effective β-blockers for improving survival after myocardial infarction. Yet the mechanisms by which carvedilol achieves this superior clinical profile are still unclear. Beyond blockade of β₁-adrenoceptors, arrestin-biased signalling via β₂-adrenoceptors is a molecular mechanism proposed to explain the survival benefits. Here, we offer an alternative mechanism to rationalize carvedilol's cellular signalling. Using primary and immortalized cells genome-edited by CRISPR/Cas9 to lack either G proteins or arrestins; and combining biological, biochemical, and signalling assays with molecular dynamics simulations, we demonstrate that G proteins drive all detectable carvedilol signalling through β₂ARs. Because a clear understanding of how drugs act is imperative to data interpretation in basic and clinical research, to the stratification of clinical trials or to the monitoring of drug effects on the target pathway, the mechanistic insight gained here provides a foundation for the rational development of signalling prototypes that target the β-adrenoceptor system.

Understanding the mechanism of drug action is no prerequisite for a drug to obtain approval from regulatory authorities, but a solid foundation for evidence-based decision-making along all stages of the drug discovery process[1,2]. Conversely, if the mechanism of drug action remains enigmatic, or if consensus is lacking on how precisely a drug alters the function of its protein target, the molecular basis for therapeutic efficacy will remain obscure.

Carvedilol, a widely prescribed cardiovascular medication, is a case in point. It belongs to a class of drugs known as beta receptor blockers (β-blockers), which are commonly used to treat hypertension and heart failure[3–5]. β-blockers are particularly effective in patients where myocardial damage is associated with overstimulation of β-adrenoceptors (βARs). βARs are prototypical class A G protein-coupled receptors (GPCRs), the targets for the majority of prescription drugs in clinical

[1]Molecular, Cellular and Pharmacobiology Section, Institute of Pharmaceutical Biology, University of Bonn, 53115 Bonn, Germany. [2]Research Training Group 1873, University of Bonn, 53127 Bonn, Germany. [3]InterAx Biotech AG, 5234 Villigen, Switzerland. [4]Department of Pharmaceutical Chemistry, Philipps-University of Marburg, 35032 Marburg, Germany. [5]Institute for Molecular Cell Biology, CMB-Center for Molecular Biomedicine, Jena University Hospital, Friedrich Schiller University of Jena, 07745 Jena, Germany. [6]Institute of Pharmacology and Toxicology, Jena University Hospital, Friedrich Schiller University of Jena, 07747 Jena, Germany. [7]Institute of Physiology I, Medical Faculty, University of Bonn, 53115 Bonn, Germany. [8]Department of Infection and Immunity, Luxembourg Institute of Health (LIH), L-4354 Esch-sur-Alzette, Luxembourg. [9]Institute of Metabolism and Systems Research and Centre of Membrane Proteins and Receptors (COMPARE), University of Birmingham, Birmingham B15 2TT, UK. [10]Institute of Experimental Cardiovascular Research, University Medical Center Hamburg-Eppendorf, 20246 Hamburg, Germany. [11]Graduate School of Pharmaceutical Science, Tohoku University, Sendai 980-8578, Japan. [12]7TM Antibodies GmbH, 07745 Jena, Germany. [13]Present address: Ikherma Consulting Ltd, Hitchin SG4 0TY, UK. ✉e-mail: kostenis@uni-bonn.de

use[6,7]. βARs are also among the most powerful regulators of cardiac function[8], and the primary targets for the endogenous catecholamines adrenaline and noradrenaline, which act as neurotransmitters and hormones to control heart rate and blood pressure[9]. β-blockers oppose the stimulating effects of catecholamines, which explains their efficacy in the clinic to reduce heart rate and contractility and, consequently, lower the cardiac oxygen demand.

From a pharmacological perspective, β-blockers are highly heterogeneous as they differ in their β₁-(cardio)selectivity[10], antioxidant[11] or vasodilative α₁AR-blocking properties[12], as well as their ability to also stimulate βARs to some extent[13,14]. This latter property was initially referred to as intrinsic sympathomimetic activity (ISA)[15] and likely rationalizes a number of clinical benefits such as less reduction in resting and maximal exercising heart rate and consequently less reduction in cardiac output, less upregulation of βARs after long-term treatment and a lower mortality rate in secondary prevention after myocardial infarction[16]. Despite the apparent benefits associated with ISA and the use of β-blockers for more than five decades, the signal transduction mechanisms underlying the low efficacy βAR activation are still unclear.

To clarify these mechanisms, we here take advantage of carvedilol, a non-selective β-blocker with ISA and additional antioxidant activity[17], as well as α₁AR and ryanodine receptor blocking properties[18–21]. We chose carvedilol for several reasons: (i) it is among the most effective β-blockers for improving survival after myocardial infarction[22,23], (ii) it is one out of four β-blockers approved for the treatment of heart failure but the only one in this subgroup with β₂-ISA[24] and (iii) it is frequently presented and used as a prototype to portray a paradigmatic signalling mechanism: G protein-independent, arrestin-dependent signalling, also known as arrestin-biased signalling[25,26]. Arrestin-biased signalling via β₂AR was even hypothesized to explain the favourable clinical profile of carvedilol[27].

In the arrestin signalling paradigm, the core signalling unit is composed of a GPCR, arrestin-2 and arrestin-3 (hereafter arr2 and arr3) and mitogen-activated protein kinases (MAPK) of the extracellular signal-regulated kinase (ERK) family but does not include heterotrimeric G proteins, the main transducers of GPCR signals[25,26]. Although recent investigations attempt to soften the strict separation between G protein and arrestin-driven signalling[28,29], the G protein versus arrestin dichotomy still remains the molecular pillar for the ligand bias paradigm[30]. The importance of arrestin-biased signalling in basic and clinical research is best exemplified by the large and steadily increasing number of >17,000 research documents (as tracked by Google Scholar) during the last two decades examining various aspects of this signalling mode in vitro and in vivo[26,28,31–46].

Intrigued by the notion that (i) endogenous bona fide arrestin-biased receptors do not evoke ERK/MAPK phosphorylation[47,48], (ii) arrestins are dispensable for the initiation of GPCR signalling to ERK[44,49] and (iii) arrestin-dependent MAPK activation requires the presence of active G proteins[35,50,51], we here chose the purported arrestin-biased carvedilol to shed more light on this enigmatic signalling mechanism.

In this work, we use well-validated cell models, genome-edited by CRISPR/Cas9, that either lack Gα proteins or arrestins, along with primary cardiomyocytes and a battery of signalling, biological and biochemical assays to propose that carvedilol induces all detectable cellular signalling via low intrinsic activation of heterotrimeric G proteins. Our investigations clarify the molecular mechanisms underlying the ISA of carvedilol at β₂ARs and have major implications for the evidence-based use of β-blockers in clinical practice. If ISA is indeed the distinctive feature that explains why some β-blockers are superior over others in heart failure clinical trials, our study resolves the conundrum as to why β-blockers with ISA prolong the life of heart failure patients.

## Results

### Carvedilol initiation of signalling downstream of β₂AR requires Gs proteins

The prevailing theory classes carvedilol as an arrestin-biased ligand at β₂AR[27], a signalling mode even proposed to explain the survival benefits for patients in heart failure clinical trials[52]. Traditionally, arrestin-biased signalling is recorded as phosphorylation of ERK MAP kinases[53]. However, all four classes of heterotrimeric G proteins, Gi/o, Gs/olf, Gq/11 and G12/13, also phosphorylate ERK[54]. Convergence of independent G protein and arrestin pathways on a common effector makes a clear-cut assignment of the upstream transducer technically difficult, if not impossible. Hence, to overcome this limitation and to clarify how carvedilol activates ERK via β₂AR, cell lines with selective deletion of either G proteins or β-arrestins are mandatory. We took advantage of CRISPR/Cas9-based HEK293 knockout cell lines, engineered to lack expression of arr2 and arr3 (hereafter Δarr2/3)[49,51], or of the three Gα subunit families Gα_{s/olf}, Gα_{q/11} and Gα_{12/13} (hereafter Δsix, reflecting deletion of six endogenous Gα proteins)[50]. Because Δsix cells still express endogenous Gα_{i/o} subunits, pre-treatment with pertussis toxin (PTX) is demanded to achieve the condition of "zero functional G" (hereafter Δsix + PTX).

To minimize inter-assay variability across our CRISPR cell lines, we stably overexpressed β₂AR at comparable cell surface abundance (Supplementary Fig. 1). We fused a SNAP-tag to the β₂AR N-terminus to visualize receptor expression and to permit direct comparison of both cellular signalling and spatiotemporal distribution within the same cellular background. The addition of the SNAP-tag did not impair ligand binding to or function of β₂AR (Supplementary Fig. 2)[55].

In agreement with active signalling[14,56–59], carvedilol-stimulated ERK phosphorylation in a transient and time-dependent manner and with lower efficacy as observed for the full synthetic agonist isoproterenol (ISO) (Fig. 1a). ERK phosphorylation was preserved in cells lacking arr2/3 (Fig. 1b), undetectable at zero functional G (Δsix + PTX, Fig. 1c) or in cells endowed with endogenous Gi/o proteins (Supplementary Fig. 3) but re-emerged with Gα_s re-expression (Fig. 1d). Seemingly, carvedilol-induced MAPK activation does not conform to an arrestin-biased signalling pattern but is, instead, genuinely initiated by Gs family proteins. In line with this notion, carvedilol-induced cell morphology changes as detected with a label-free optical biosensor, which provides real-time measures for global cell activation, were strictly Gα_s protein-driven (Fig. 1e–h and Supplementary Fig. 4). Apparently, Gs proteins play a pivotal role in relaying β₂AR activation to intracellular effectors.

Activation of Gs family proteins typically stimulates the production of the second messenger cAMP by adenylyl cyclase (AC) isoforms[60]. Indeed, cAMP abundance was enhanced by the action of carvedilol at the β₂AR, and this effect was nullified only in G protein-deficient cells yet partially recovered with Gα_s re-expression (Fig. 1i–l and Supplementary Fig. 5). Thus, the entire set of cellular responses evoked with carvedilol showed equivalent dependence on Gs. Arrestins, on the contrary, were dispensable for initiation of all downstream signals, even those transmitted to pERK, the hallmark feature of arrestin-biased signalling[25–27,53].

### Carvedilol-occupied β₂ARs recruit and stimulate Gs

Gs dependence on carvedilol signalling led us to investigate its capacity to promote the transition of β₂AR into an active state conformation. To this end, we used miniGs, a G protein surrogate composed of the engineered GTPase domain of Gα_s, as a conformational sensor for the active receptor state[61–63]. We probed β₂AR miniGs interaction with an assay based on enzyme complementation between split fragments of NanoLuc® luciferase fused to the β₂AR and miniGs, respectively (NanoBiT assay principle in Fig. 2a). Cells transfected with the split sensor pair led to detectable increases in luminescence signals upon

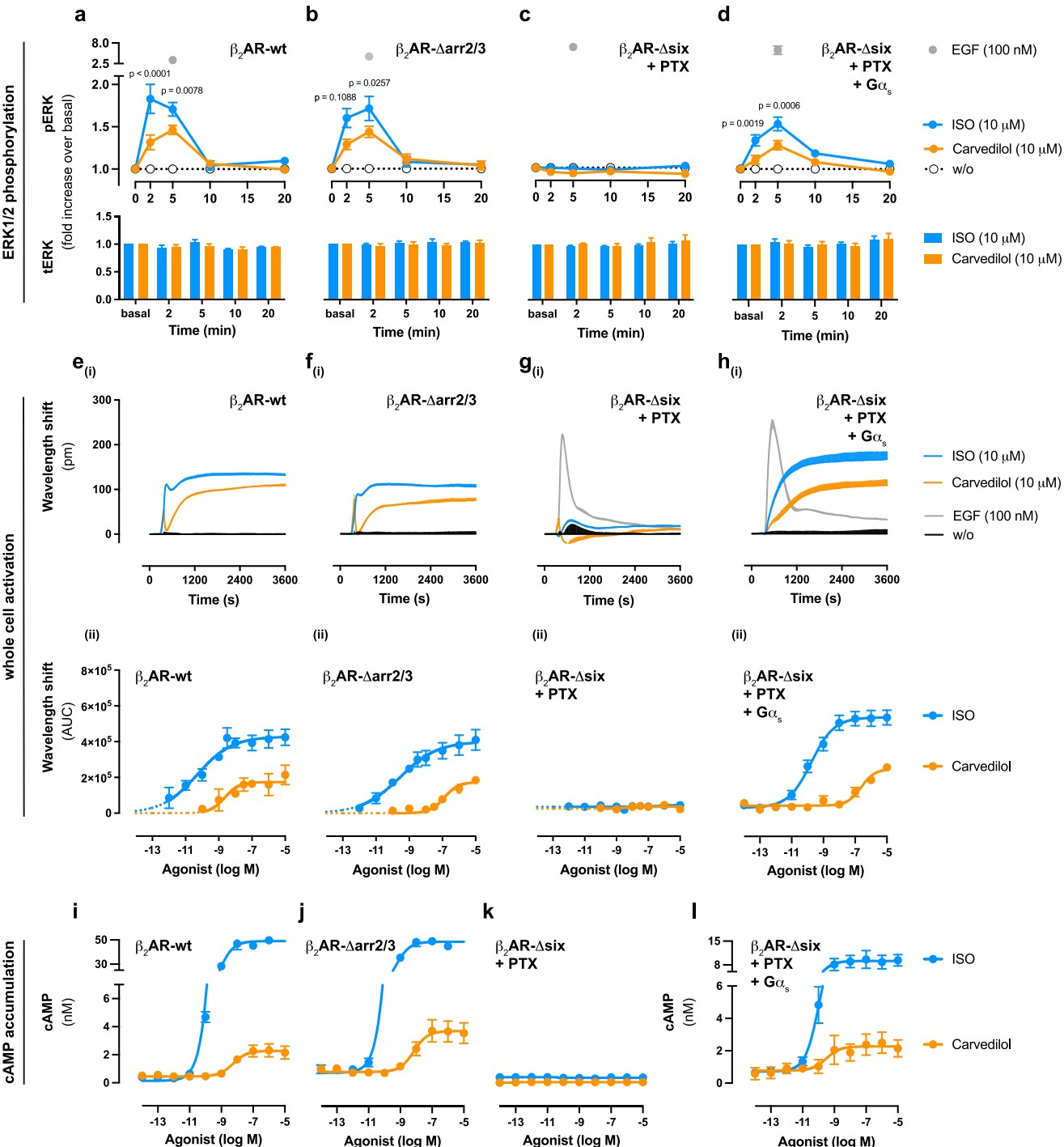

**Fig. 1 | Carvedilol initiation of signalling downstream of β₂AR requires Gs proteins.** ERK1/2 phosphorylation, cell morphological changes and cAMP accumulation in wild-type (wt) and genome-edited HEK293 cells, stably expressing SNAP-tagged β₂ARs, upon stimulation with either carvedilol or Isoproterenol (ISO). **a–d** Temporal pattern of ERK1/2 phosphorylation and total ERK1/2 in **a** wild-type, **b** Δarr2/3, **c, d** Δsix cells pre-treated with PTX in the absence (**c**) and presence (**d**) of re-expressed heterotrimeric G protein subunit Gα_s after stimulation with the indicated ligands. **e–h** Representative optical recordings of cell morphological changes (**e_i–h_i**) and corresponding concentration-response curves (**e_ii–h_ii**) upon stimulation with carvedilol, ISO or epidermal growth factor (EGF) as viability control. **i–l** Quantification of carvedilol- or ISO-induced cAMP accumulation in **i** wild-type, **j** Δarr2/3 and **k, l** Δsix cells pre-treated with PTX in the absence (**k**) and after re-expression of Gα_s (**l**). Kinetic traces in **e_i–h_i** of a representative experiment (from 3–9 independent experiments) are shown as means + SD, measured in triplicate. Summarized data are depicted as means +/± SEM (**a, b**: $n = 6$; **c, d**: $n = 9$; **e_ii–g_ii**: $n = 3$; **h_ii**: $n = 9$; **i–k**: $n = 3–5$ per ligand; **l**: $n = 3–4$ per ligand). **a–d**: two-way ANOVA followed by Šídák's post hoc test. pm picometre, w/o without. Source data are provided as a Source Data file.

carvedilol addition and revealed slower kinetics as well as lower maximal amplitudes than those elicited by ISO (Fig. 2a and Supplementary Fig. 6). No such effect for carvedilol was observed in cells transfected with β₂AR and miniGi (Supplementary Fig. 7), in agreement with the absence of detectable Gi/o signalling for carvedilol in both pERK and

whole cell activation assays. β₂AR-Gs interaction was confirmed when we altered the labelling strategy to detect bioluminescence resonance energy transfer (BRET) between NanoLuc®-tagged β₂AR and Venus-tagged miniGs (Supplementary Fig. 8). From these data, we concluded that carvedilol-induced proximity between β₂AR and miniGs, albeit

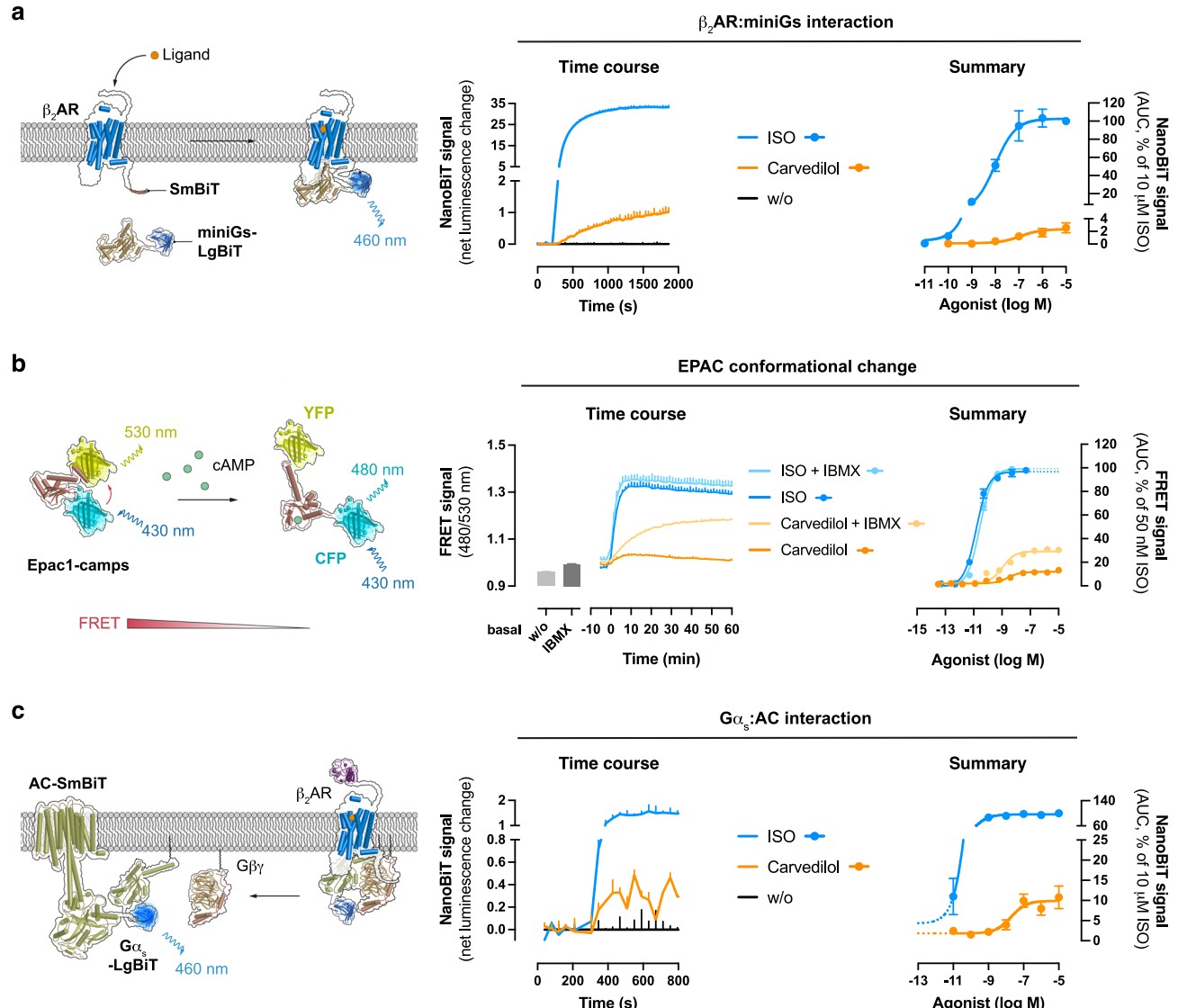

**Fig. 2 | Recruitment and activation of Gs by carvedilol-occupied β₂ARs.**
**a** Schematic of the NanoBiT complementation assay between ligand-activated β₂AR C-terminally tagged with the small fragment (SmBiT) of Nanoluciferase (NanoLuc®) and miniGs N-terminally fused to the large fragment of NanoLuc® (LgBiT). Ligand application to the transfected cells induces proximity between the labelled proteins and thus, complementation of a functional NanoLuc® enzyme. Shown are representative kinetic traces of NanoBiT complementation in HEK293 wt cells transiently transfected with β₂AR-SmBiT and LgBiT-miniGs after treatment with either 10 μM carvedilol or 10 μM ISO as well as concentration-response curves derived therefrom. **b** Assay principle underlying detection of cAMP as a FRET decrease with the cytosolic Epac1-camps sensor. Data shown are representative FRET recordings in SNAP-β₂AR-HEK293 wt cells after addition of either carvedilol (10 μM) or ISO (50 nM) in the presence or absence of the pan-PDE inhibitor IBMX (500 μM) and corresponding concentration-response curves. FRET recordings of cAMP dynamics were performed in the subsaturating sensor range for carvedilol but at sensor saturation for ISO. **c** Cartoon of NanoBiT-based detection of interactions between Gαs-LgBiT and AC5-SmBiT, accompanied by representative NanoBiT time courses after treatment of cells with either 10 μM carvedilol or 10 μM ISO and corresponding concentration-effect relationships. Kinetic traces of a representative experiment (from 3–4 independent experiments) are shown as means + SD of 2–4 technical replicates. Summarized data are presented as means ± SEM (**a**: $n = 3$; **b**: $n = 4$; **c**: $n = 3$). w/o without. Source data are provided as a Source Data file.

slower in onset and with reduced maximal amplitude as compared to the effect evoked by ISO.

Because Nanoluciferase complementation assays measure effector recruitment but not signalling, we monitored ligand-mediated dynamic changes of cAMP utilizing an EPAC-based FRET biosensor (schematic of sensor in Fig. 2b). Temporal resolution of cAMP abundance as net result of its production and breakdown revealed concentration-dependent enhancement by carvedilol, slower and less efficacious as the effect induced by ISO (Fig. 2b). Carvedilol-induced cAMP formation was further amplified when its breakdown by phosphodiesterases (PDEs) was specifically interdicted with the pan-PDE inhibitor IBMX (Fig. 2b).

Apparently, the cellular cAMP pool, elevated by carvedilol via β₂ARs, is tightly controlled by active PDEs.

Cytosolic cAMP levels result from a fine balance of second messenger formation by AC isoforms and degradation by PDEs. Entirely consistent with Gs recruitment (Fig. 2a and Supplementary Fig. 8) and signalling (Fig. 2b), carvedilol also promoted molecular proximity between Gαs and AC as read out by NanoLuc®-based enzyme complementation between Gαs fused to the large and AC5 fused to the small fragment of Nanoluciferase, respectively (Fig. 2c and Supplementary Fig. 9). From these data we concluded that carvedilol exerts its detectable cellular signals by activation of the β₂AR-Gs-AC-cAMP signalling axis.

## Carvedilol elevates cAMP and spontaneous beating in primary cardiomyocytes

Cardiomyocyte cAMP synthesis after β-adrenergic activation is a fundamental physiological process in the cardiovascular system that impacts beat-to-beat heart contraction and relaxation in order to adjust cardiac output to the oxygen demand of the body[64]. It is, therefore, imperative to investigate whether carvedilol also elevates cAMP in cardiomyocytes, where βARs are endogenously expressed. We used the genetically encoded plasma membrane-localized Epac-based FRET sensor pmEpac1-camps to read out cAMP dynamics in adult murine ventricular myocytes (Fig. 3a)[65]. Consistent with our previous findings in $\beta_2$AR-HEK293 cells (cf. Fig. 2b), treatment with ISO or carvedilol alone led to detectable increases in cAMP levels and their effects were further enhanced when forskolin, a direct AC activator, was coapplied with IBMX to define the maximum possible FRET change for each cardiomyocyte (Fig. 3b). Pre-treatment of cells with ICI-118,551 (ICI), a $\beta_2$-selective antagonist, abolished carvedilol-induced cAMP formation, while pre-treatment with CGP-20712A (CGP), a $\beta_1$-AR selective inhibitor, had no effect (Fig. 3b, c). We concluded that carvedilol enhances cAMP formation in primary adult cardiomyocytes by specific activation of $\beta_2$AR.

To investigate whether carvedilol-induced cAMP elevation translates into speeding up of spontaneous cardiomyocyte beating, we utilized the label-free, impedance-based CardioExcyte96 detection platform, which permits non-invasive recording of beating frequency in real-time[66]. Congruent with the notion that $\beta_1$ and $\beta_2$ARs contribute to contractile responses of both embryonic and adult cardiomyocytes[67], ISO, a mixed $\beta_1$ and $\beta_2$AR agonist as well as carvedilol, which elevates cAMP via $\beta_2$AR, measurably increased the beating frequency of neonatal mouse ventricular myocytes (Fig. 3d). ISO-enhancement of beating frequency was countered by pre-treatment with the $\beta_2$-preferring ICI and the $\beta_1$-preferring CGP antagonists (Fig. 3e, f). Because $\beta_1$AR has a larger impact on β-adrenergic control of spontaneous cardiomyocyte beating (Supplementary Fig. 10), and because the predominant action of carvedilol is blockage of $\beta_1$AR, we reasoned that $\beta_1$ inhibition by carvedilol might mask its own low efficacy $\beta_2$ activation[68]. Indeed, carvedilol enhancement of spontaneous beating was small but significant, unmasked in the presence of CGP ($\beta_2$AR stimulation permitted), and fully reversed by ICI ($\beta_2$AR stimulation countered) (Fig. 3g, h). We concluded that the detectable positive signals of carvedilol in cardiomyocytes originate from specific activation of the $\beta_2$AR-Gs-AC-cAMP axis. Thus, in addition to limiting the effects of catecholamine excess (classical $\beta_1$ blockade), carvedilol activates $\beta_2$AR and thereby provides some background sympathetic activity via low efficacy G protein signalling. Given that $\beta_1$ but not $\beta_2$ARs are downregulated in heart failure[69–72] along with $\beta_2$AR redistribution from the T-tubules to the cell crest, giving rise to cell-wide cAMP propagation in the failing cardiomyocyte[73], it is conceivable that low efficacy $\beta_2$AR activation even occurs in human heart failure patients and may contribute to the positive effect of carvedilol.

## Carvedilol does not promote detectable internalization of $\beta_2$AR

Of the possible mechanisms by which carvedilol could drive its activating signals (G proteins versus arrestins or even as yet unidentified effectors), low efficacy G protein signalling is clearly the most likely. Nevertheless, carvedilol may still be unique among β-blockers in that low efficacy G protein signalling is linked to disproportionately large desensitization. Because this assumption has not been directly tested yet, we investigated carvedilol-stimulated internalization of $\beta_2$AR. In agreement with earlier findings[74], ligand-stimulated internalization of $\beta_2$AR in HEK293 cells is an arrestin-dependent process (Supplementary Fig. 11). However, to our surprise, and unlike previously proposed[27] carvedilol did not visibly alter $\beta_2$AR redistribution in living $\beta_2$AR-

HEK293 cells in contrast to the robust endocytosis achieved with ISO (Fig. 4a and Supplementary Fig. 12). Line scan analysis confirmed $\beta_2$AR confinement to the plasma membrane in the absence of agonist, as shown by the two sharp fluorescence peaks flanking the low fluorescence cytoplasmic area (Fig. 4b). Carvedilol did not alter fluorescence intensity distribution in contrast to ISO, which dampened plasma membrane but enhanced cytoplasmic fluorescence (Fig. 4b). Apparently, the conformational change produced by carvedilol does not promote detectable $\beta_2$AR internalization, contrary to a previous study[27], but in agreement with other works[75,76].

Whilst fluorescence microscopy is well-suited to qualitatively examine how GPCR ligands alter receptor location in living cells[77,78], the effects produced by low efficacy ligands may be transparent to this technique. We, therefore, quantified in real-time the loss of surface $\beta_2$AR using the sensitive diffusion-enhanced resonance energy transfer (DERET) technology (Fig. 4c)[79–82]. Carvedilol induced no observable internalization of plasma membrane-localized $\beta_2$AR over time in contrast to the rapid and pronounced receptor loss evoked by ISO (Fig. 4d).

GPCRs are highly mobile but become immobilized in clathrin-coated pits (CCPs) prior to their removal from the cell surface[83]. We used a single-particle tracking analysis to study $\beta_2$AR trapping in CCPs to understand why carvedilol does not visibly internalize $\beta_2$AR in living cells. To this end, we simultaneously imaged the plasma membrane diffusion of $\beta_2$AR and clathrin, fused to a SNAP-tag and GFP, respectively[84], using fast two-colour single-molecule microscopy combined with single-particle tracking[55]. Carvedilol did not significantly alter $\beta_2$AR lateral mobility when compared to the effects induced by solvent and did not significantly promote the occurrence of $\beta_2$ARs in CCPs (Fig. 4e, f). ISO-stimulated receptors, in contrast, rapidly accumulated and became trapped in CCPs (Fig. 4e, f), consistent with altered diffusional mobility of the entire receptor population (Fig. 4g). This can be assigned to receptor trapping because the diffusion coefficient of the freely diffusing population was largely unaffected by any treatment regimen (Fig. 4g). Viewed collectively, three complementary methods indicate that carvedilol produces a $\beta_2$AR conformation that is signalling competent, yet unable to undergo detectable internalization.

## Carvedilol-bound $\beta_2$AR is a poor target for arrestin binding

After activation, GPCRs localize to CCPs via their interaction with arrestins and other endocytic proteins such as adapter protein 2 (AP-2)[85]. Because CCP trapping requires prior arrestin recruitment[86], we speculated that enrichment of cells with exogenous arrestins and possibly G protein-coupled receptor kinases (GRKs) to enhance the affinity of arrestin for the receptor[87–89], might boost carvedilol-dependent internalization. However, overexpression of GRK2, arr3 or both in $\beta_2$AR-HEK293 cells failed to promote measurable receptor internalization in response to carvedilol, while the same transfection regime effectively enhanced the effect of ISO (Fig. 5a). Clearly observable $\beta_2$AR internalization for ISO but not carvedilol was also evident from line-scans of the acquired images (Fig. 5b) and confirmed by DERET-based quantification of $\beta_2$AR surface loss (Fig. 5c, d). These data suggest that the lack of measurable $\beta_2$AR internalization in our $\beta_2$AR-HEK293 cells is not due to an insufficient abundance of endocytosis adaptors and cannot be enforced even if these are overexpressed.

Therefore, we speculated that carvedilol-occupied $\beta_2$AR may be a poor target for arrestin binding. To test this assumption, we determined ligand-induced arrestin recruitment to $\beta_2$AR using a number of complementary protein-protein interaction assays. Regardless of whether we used split Nanoluciferase (NanoBiT-based) enzyme complementation between $\beta_2$AR fused to the small ($\beta_2$AR-SmBiT) and arrestin-2/3 fused to the large NanoLuc® fragment (LgBiT-arr2/3) (Fig. 5e, f and Supplementary Fig. 13), $\beta_2$AR-NanoLuc® in combination with Venus-arr3 (Supplementary Fig. 14), or $\beta_2$AR-SmBiT along with AP-

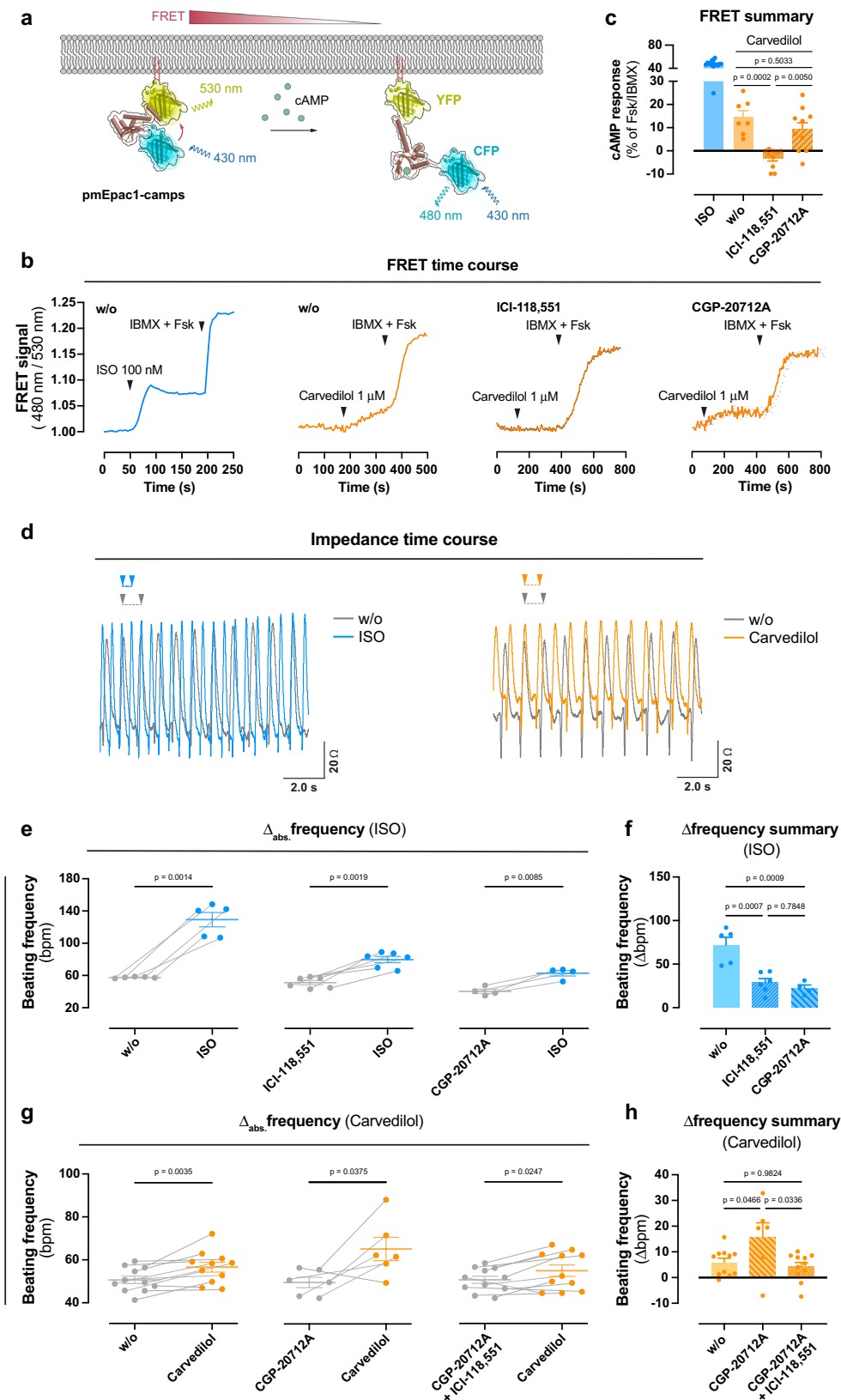

**a** FRET

pmEpac1-camps

cAMP

YFP 530 nm / CFP 480 nm 430 nm

**b** FRET time course

**c** FRET summary

Carvedilol

**d** Impedance time course

**e** Δabs. frequency (ISO)

**f** Δfrequency summary (ISO)

**g** Δabs. frequency (Carvedilol)

**h** Δfrequency summary (Carvedilol)

2 nonbinding versions of LgBiT-arr2/3 to extend the lifetime of the luminescence signal (Fig. 5g and Supplementary Fig. 15)[86,90], β$_2$ARs recruited neither arrestin isoform after carvedilol addition despite exogenous expression of all components.

A notable difference between our data and the original report is that experimental evidence supporting arrestin recruitment by carvedilol-activated β$_2$AR was collected with a β$_2$AR chimera, wherein the C-tail has been swapped with that of the vasopressin V2 receptor (β$_2$V$_2$)[27]. This modification converts β$_2$AR from a transient (class A) to a stable (class B) arrestin-binder[91] to produce long-lived receptor arrestin complexes[92,93]. To our surprise, in our hands even carvedilol-bound β$_2$V$_2$ did not recruit arrestins in two complementary BRET approaches, using

**Fig. 3 | Carvedilol elevates cAMP and spontaneous beating in primary cardiomyocytes. a** Assay design for cAMP detection in adult mouse ventricular myocytes (AMVM) using the plasma membrane-bound Epac1-camps sensor (pmEpac1-camps). **b** Representative ligand-mediated kinetic pmEpac1-camps recordings in AMVM single cells without and after pre-treatment with the $\beta_2AR$ selective blocker ICI-118,551 (ICI, 1 µM) or the $\beta_1AR$ selective blocker CGP-20712A (CGP, 1 µM). **c** Summarized ligand-mediated cAMP response of AMVM single cells treated with ISO ($n = 20$) or carvedilol without ($n = 7$) or with ICI ($n = 10$) or CGP ($n = 10$) relative to the effect achieved with pan-PDE inhibitor IBMX (100 µM) in combination with AC activator Fsk (10 µM). **d** Representative impedance signals from spontaneously beating neonatal mouse ventricular cardiomyocytes before (w/o) and after application of 300 nM ISO (blue, left) or 300 nM Carvedilol (orange, right). **e, f** Effect of ISO in control conditions (w/o, $n = 5$) or in the presence of 10 µM ICI ($n = 6$) or 10 µM CGP ($n = 4$) on absolute beating frequency. **g, h** Effect of carvedilol in control conditions (w/o, $n = 11$), in the presence of CGP ($n = 6$) or in the presence of both CGP and ICI (10 µM each, $n = 11$) on absolute beating frequency. Kinetic recordings of single-cell FRET microscopy in **b** and beating frequency analyses in **d, e, g** are from single cells and cell populations, respectively, obtained from three different animal preparations. Only cardiomyocytes with spontaneous beating ≤60 bpm were considered for the impedance time course analysis in **e–h**. Data in **c, e–h** are presented as means ± SEM. **e, g**: two-tailed paired Student's $t$-test; **c, f, h**: one-way ANOVA with Tukey's post hoc test. Source data are provided as a Source Data file.

pairs of $\beta_2V_2$ C-terminally fused to EYFP and Renilla luciferase fused to arrestin-3 (RLuc-arr3) as well as $\beta_2V_2$ fused to NanoLuc® and Halo-Tag-labelled arr3 (Fig. 5h and Supplementary Fig. 16). Consistent with the lack of arrestin recruitment, carvedilol neither promoted appreciable internalization of $\beta_2V_2$ (Supplementary Fig. 17a, b) nor diminished its surface abundance (Supplementary Fig. 17c). ISO, in contrast, enabled robust arrestin recruitment irrespective of the BRET labelling strategy (Fig. 5h), and effectively promoted $\beta_2V_2$ internalization and surface loss (Supplementary Fig. 17). As a caveat to our posit, it remains possible that carvedilol is a very slow and low efficacy agonist also for arrestin recruitment and internalization, but that current assays may lack sensitivity and/or timescale to detect this low efficacy at the maximum possible concentration. Collectively, our complementary experimental evidence strongly suggests that carvedilol cannot be considered arrestin-biased because it does not demand arrestins for initiation of signalling or desensitization purposes.

### Ligand-specific $\beta_2AR$ phosphorylation and unbiased molecular dynamics simulations fully explain carvedilol's signalling profile

Interaction with arrestins requires the receptor to be phosphorylated at multiple sites by various GRK isoforms[32,94], and this, in turn, necessitates ligand-induced transition to a particular active conformation[95]. Puzzled by the absence of arrestin recruitment to $\beta_2V_2$, we speculated that carvedilol may be unable to produce this particular active state. We tested our assumption with a conformation-specific nanobody (Nb80), generated previously to sense the fully active $\beta_2AR$[96]. As expected, ISO robustly recruited Nb80 to $\beta_2AR$ as determined by NanoLuc® complementation between $\beta_2AR$-smBiT and Nb80-LgBiT (Fig. 6a and Supplementary Fig. 18). However, contrary to miniGs which was recruited by carvedilol to $\beta_2AR$ (cf. Fig. 2a), Nb80 was not (Fig. 6a and Supplementary Fig. 18). Apparently, the signalling competent conformation stabilized by carvedilol remains unnoticed by Nb80, consistent with the earlier finding that Nb80 does not shift the affinity of carvedilol for the $\beta_2AR$[97].

To capture the differences between ISO- and carvedilol-liganded $\beta_2AR$ in atomic detail, we used metadynamics molecular dynamics simulations. Within the measure to track activation[98], we find that carvedilol stabilizes a conformation that is distinct from both the ISO-induced active state and the unliganded receptor in the apo-state (Fig. 6b). ISO stabilizes a clearly active state, signified by the preservation of the second intracellular loop ICL2 as an α-helix (Fig. 6c), and often accompanied by the formation of a hydrogen bond between Y141$^{ICL2}$ and D130$^{3.49}$ of the highly conserved DRY motif (Fig. 6c)[99]. Carvedilol, in contrast, does not stabilize the same active state but another conformation with molecular features compatible with partial Gs protein activation (Fig. 6c and Supplementary Methods to provide more detailed technical descriptions).

Complementary to Nb80 and MD simulations, we used the phosphorylation-independent arrestins arr3-R170E and arr3-F388A as conformational reporters for active state receptors. Consistent with their phosphorylation-independence[100–103], arr3-R170E and arr3-F388A were effectively recruited to $\beta_2AR$ by ISO, both in the presence and

collective genetic absence of GRK2/3/5/6[88] (Fig. 6d, e). Carvedilol, in contrast, was completely ineffective (Fig. 6d, e), which is why we concluded that its conformational adjustment of the intracellular receptor face suffices exclusively for initiation of low efficacy Gs signalling but neither for recognition by active state conformational biosensors nor by endocytic adaptors. We surmised that carvedilol-liganded $\beta_2AR$ may be no or, alternatively, an inherently poor target for GRK phosphorylation and arrestin binding.

The phosphorylation barcode hypothesis states that ligand-specific phosphorylation patterns direct distinct functional outcomes[104]. Because carvedilol signals via the Gs-cAMP pathway in the absence of detectable desensitization, we used phosphosite-specific antibodies to probe $\beta_2AR$ phosphorylation at both PKA and GRK sites (Fig. 6f). Entirely consistent with all our earlier data, carvedilol-stimulated phosphorylation of $\beta_2AR$ at position S261, a known PKA site but not at positions S355/S356 and T360/S364, two well-established GRK sites[105–108], even in the presence of overexpressed GRK2 and 6 (Fig. 6g). ISO, in contrast, promoted robust phosphorylation at all sites, entirely consistent with Gs signalling and subsequent GRK-mediated desensitization (Fig. 6g)[109]. Because PKA site phosphorylation does not result in arrestin recruitment[110,111], our site-specific phosphorylation analyses reconcile both the positive cellular signals induced by carvedilol via the Gs-AC-cAMP axis and the resistance of $\beta_2AR$ to carvedilol-induced internalization/desensitization.

Viewed collectively, our study calls into question the dogmatic classification of carvedilol as an arrestin-biased ligand[27], but provides, instead, the relevant mechanistic framework to reclassify the mode of action of one of the most popular beta-blockers, and how it induces its positive signals in living cells.

## Discussion

β-blockers are widely prescribed medications to treat cardiovascular diseases such as hypertension and heart failure[3–5]. Yet, despite their introduction into clinical practice over five decades ago and their widespread clinical use, the mechanisms by which β-blockers achieve favourable clinical profiles are still not clearly defined. If mechanisms transform into explanations for clinical phenotypes, i.e. become associated with survival benefits for patients, then delineating such enabling mechanisms should be mandatory if not imperative. This was the goal of the present study, and the uncovered mechanism that carvedilol exerts its positive signalling through $\beta_2AR$ via low efficacy G protein activation is our major finding.

Akin to all βAR-blocking agents, the predominant action of carvedilol is to counter the stimulating effects of endogenous catecholamines by antagonizing $\beta_1$ and $\beta_2ARs$, but it also possesses some residual agonistic activity, proposed to explain its uniqueness over other β-blockers in heart failure clinical trials[27]. The question of how carvedilol achieves this positive signalling has puzzled scientists for decades, but, despite this, the enabling mechanism is still undefined. Our study resolves this conundrum and clarifies, unambiguously, that the unique signalling activity of carvedilol at $\beta_2AR$ arises from low efficacy activation of heterotrimeric Gs proteins but does not require

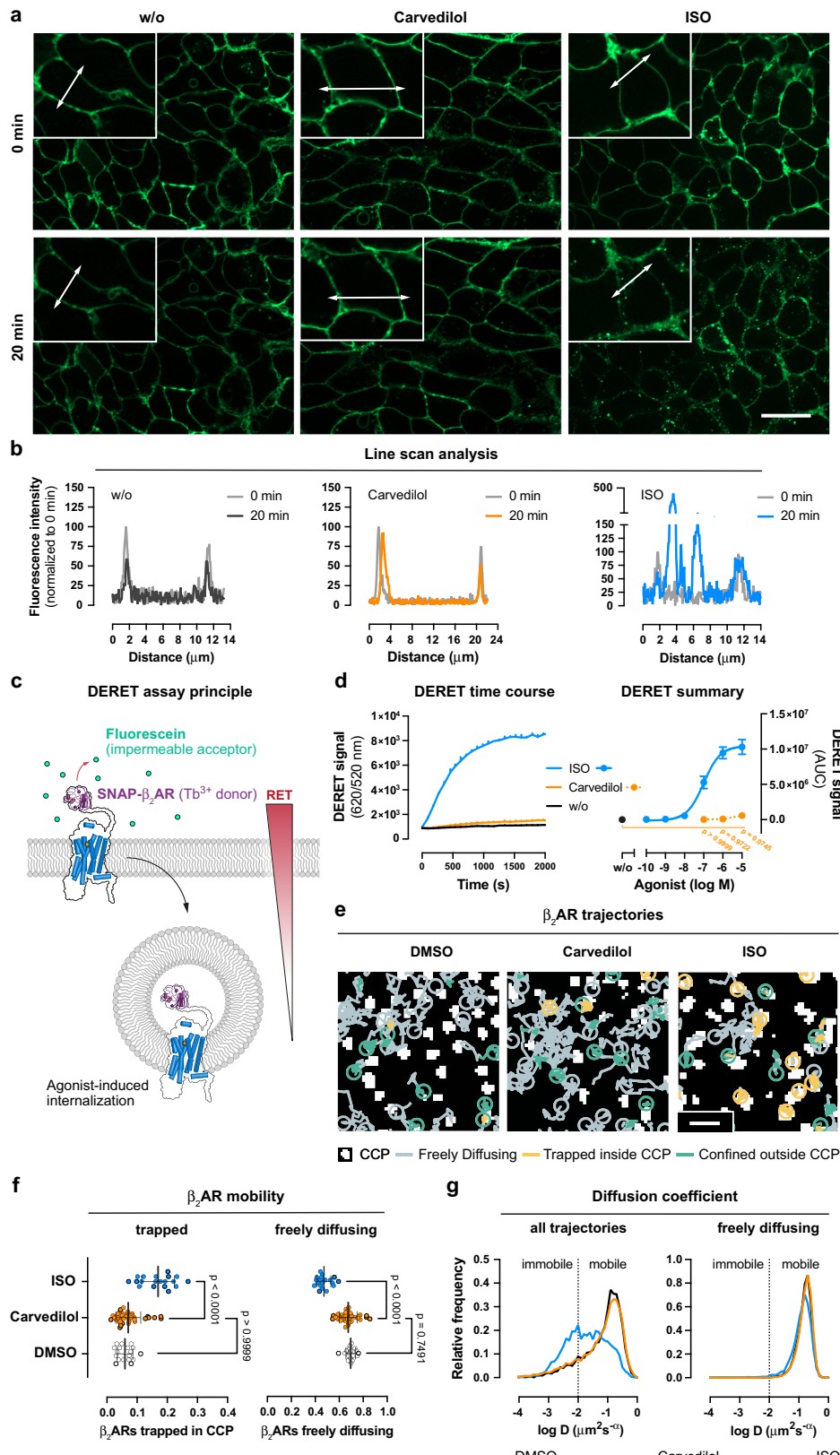

contribution from arrestins. Our results are mechanistically consistent with earlier findings obtained with carvedilol on human explants from adult ventricular myocardium[24] and on membranes from a transgenic rat model for left ventricular pressure overload[72] detecting GTP-modifiable binding, a hallmark feature of G protein activation. They

also agree with independent investigations showing enhancement by carvedilol of bulk cytosolic cAMP[14,68], of membrane-delimited and spatially restricted cAMP as detected with FRET-based cAMP reporters[56], and of cAMP-induced transcriptional changes as determined in CRE-reporter gene assays[57]. Notably, the measured cAMP

**Fig. 4 | β2ARs do not internalize after carvedilol addition. a** Structured-illumination micrographs (representative of three independent experiments) of HEK293 wt cells stably expressing SNAP-tagged β2ARs treated with buffer (w/o), 10 μM carvedilol or 10 μM ISO for 20 min using focus control. **b** Line scan analysis along a trajectory, indicated by double-faced arrows in **a**, of buffer-, carvedilol- or ISO-mediated changes in fluorescence intensity. **c** Schematic illustrating the DERET assay principle displaying how the terbium donor allows resonance energy transfer (RET) to occur with fluorescein acceptor molecules. Agonist-induced receptor internalization decreases the RET to the cell impermeable fluorescein acceptor and is visible as upward deflected trace over time. **d** Representative kinetic recording of ligand-induced DERET in wt SNAP-β2AR-HEK293, treated either with 10 μM carvedilol or 10 μM ISO, and concentration-response curves derived from three independent experiments. **e** CCP trapping analysis showing representative overlays of β2AR trajectories and the CCP mask. The coloured traces indicate trajectories of ligand-stimulated β2ARs relative to clathrin-coated structures, imaged with TIRF microscopy in CHO cells. Trajectory portions are coloured according to whether they are detected as free (grey), trapped in CCPs (yellow) or confined outside CCPs (green). **f, g** Proportion of trapped or freely diffusing receptors per cell (each dot is averaged data from one cell) (**f**), as well as the empirical probability density estimated for all or only the freely diffusing trajectory portions of the generalized diffusion coefficient D (**g**). **d, f:** One-way ANOVA with Kruskal–Wallis post hoc test. Summarized data is shown as mean ± SEM of three independent experiments. Scale bars are 20 μm for **a** and 1 μm for **e**, respectively. Source data are provided as a Source Data file.

elevations were small when detected as bulk cAMP in the cytosol[14,56], larger when assessed with spatially restricted cAMP sensors close to the plasma membrane[56], and amplified even further when measured at the level of gene transcription[14]. If cAMP is the messenger produced by the action of carvedilol at the β2AR, the signalling components typically involved are Gs proteins, which subsequently activate adenylyl cyclases to convert ATP to cAMP. Indeed, our study showed that carvedilol-mediated cAMP production was abrogated in cells genome-edited by CRISPR/Cas9 to lack all functional Gαs alleles, but re-emerged when Gαs was introduced by transient transfection. Intriguingly, carvedilol-mediated ERK phosphorylation, the hallmark feature of arrestin-biased signalling, showed an equivalent dependence on Gs, while arrestins were dispensable to elicit positive ERK signals.

Therefore, the mechanism proposed here is distinct from arrestin-biased signalling, which was proposed for carvedilol in earlier studies[27]. Yet, it is entirely consistent with the notion that low efficacy G protein activation does not trigger desensitization and is, therefore, not associated with arrestin recruitment, as shown herein and previously by others[112–114]. Notably, carvedilol is being applied in preclinical research to implicate arrestin-biased signalling in diverse experimental paradigms[33,34]. These results call for a more cautious interpretation in this regard because the outcome of the present study clearly puts forward an alternative scenario to account for the positive cellular signalling of carvedilol: low efficacy activation of heterotrimeric G proteins rather than arrestin-biased signalling.

Arrestin-biased signalling has been a recurring theme of β-blocker action proposed not only for carvedilol but also for additional β-blockers with agonist activity, such as the β2-selective ICI-118,551 or the non-selective propranolol[115]. In defence of the authors of these previous studies, the claims were made when it was technically impossible to eliminate all relevant G proteins in the applied cellular systems[116,117]. We speculate that these other β2AR ligands, hypothesized to signal through β-arrestins[115,118] or an alternative G protein-independent mechanism[14], likely adopt the same signalling mode as shown herein for carvedilol, i.e. transmit their positive signals by low efficacy G protein activation. However, unlike carvedilol, propranolol is not approved for the treatment of heart failure. This is surprising considering the significant reduction in mortality achieved with propranolol in clinical studies of heart failure patients with prior myocardial infarction[119,120]. The new insight into carvedilol signalling provided herein may therefore herald the comeback of molecular mechanisms, initially defined as ISA[15] but subsequently confounded with arrestin-biased signalling[27]. Carvedilol is still classed as arrestin-biased ligand[33,34,121]. We propose to reclassify this ligand and denote its positive signalling as partial G protein agonism.

One caveat that deserves specific mention is that the majority of assays to disentangle G protein versus arrestin-biased signalling were performed in overexpression systems. Hence, extrapolation of our data to primary cells or the in vivo situation must be performed with caution and highlight the need to re-investigate the mechanism by which carvedilol exerts its positive signals once comparable techniques become available for primary cells or even whole animals.

Notwithstanding these caveats, our new knowledge adds value to the future use of β-blockers in humans: Insight into how precisely drugs alter the function of their molecular targets is the foundation to inform the design of screens, to develop biological assays that visualize the proposed mechanism, to identify suitable disease models, to stratify patients for clinical trials, to identify biomarkers for therapeutic efficacy on the target pathway even in patients and, finally, to conceptualize improved or even novel types of medicines with superior efficacy and benefit for patients. β-blockers are a class of medicines, which are widely used to treat hypertension, to protect against recurrent heart attacks and hence to prolong the life of heart failure patients[3–5]. If ISA does indeed explain why some β-blockers are superior to others in the treatment of heart failure, our study solves the enigma of why β-blockers with ISA prolong the life of heart failure patients.

## Methods

### Chemical reagents and antibodies

Coelenterazine was purchased from Carbosynth. Linear polyethylenimine (PEI, 25 kDa) was supplied by Polyscience. FuGENE® HD Transfection Reagent, furimazine (NanoGlo®), HaloTag® NanoBRET™ 618 ligand and T4 DNA ligase were from Promega. Phosphosite-specific antibodies pS355/pS356-β2 (Cat. no.: 7TM0029A) and pT360/pS364-β2 (Cat. no.: 7TM0029B) were from 7TM Antibodies, and the antibody against pS261 (Cat. no.: PA5-12977) was from Invitrogen. Anti-HA-tag antibodies were purchased from Cell Signaling Technology. Hanks balanced salt solution (HBSS), Dulbecco's modified Eagles's medium (DMEM), Lipofectamin 2000, HA-beads, TetraSpeck fluorescent beads, 96-well white polystyrene LumiNunc microplates, all antibiotics and pertussis toxin (PTX) were from Thermo Fisher. The HTRF cAMP accumulation kit and the DERET substrate, SNAP-Lumi4-Tb, were purchased from Cisbio. SNAP-Surface Alexa Flour 488, SNAP-Surface Alexa Flour 549, SNAP-Surface Alexa Flour 649, Q5® High-Fidelity polymerase, NotI-HF®, ApaI, anti-SNAP-tag antibody (Cat. no.: P9310S) were obtained from New England Biolabs (NEB). Effectene was from Qiagen. All other chemicals and compounds were from Sigma Aldrich. TopSeal-A PLUS, Dihydroalprenolol Hydrochloride ([³H]-DHA, 250μCi, 9.25MBq), WGA PVT 500 MG SPA Beads, and OptiPlate-96, and White Opaque 96-well Microplates were from Perkin Elmer.

### Plasmids

The pSNAP-β2AR plasmid was obtained from Cisbio. HA-tagged β2AR in pcDNA3.1 was from www.cdna.org. Flag-β2AR-YFP and Flag-β2V2-YFP were provided by Cornelius Krasel (Department of Pharmacology and Toxicology, Philips-University Marburg, Germany)[122]. C-terminally NanoLuc-tagged human β2AR (β2AR-NanoLuc), N-terminally Venus-tagged MiniGαs and N-terminally Venus-tagged arr3 were kindly

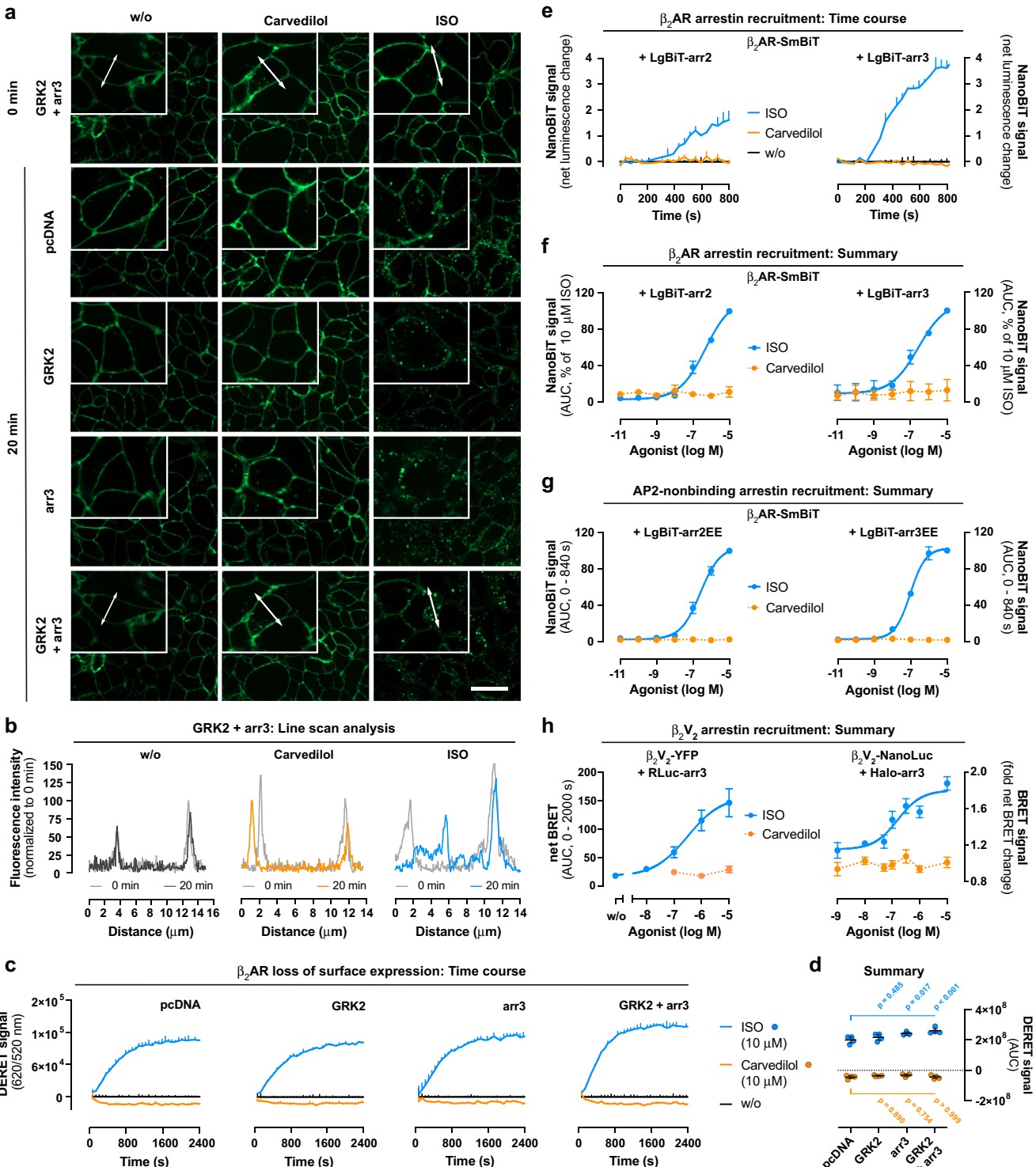

**Fig. 5 | Carvedilol-liganded β₂AR is a poor target for arrestin binding.**
**a** Structured-illumination micrographs (representative of three independent experiments) of SNAP-β₂AR-HEK293 wt cells transiently transfected with either GRK2, arr3 or both prior to and 20 min after addition of solvent control (w/o), 10 μM carvedilol or 10 μM ISO. **b** Line scans along a trajectory, indicated by double-faced arrows, corresponding to the panels in **a** showing fluorescence intensity distribution of cells transfected with GRK2 and arr3 after treatment with solvent control, carvedilol and ISO. **c, d** Representative DERET measurements in SNAP-β₂AR-HEK293 wt cells transiently transfected with pcDNA (*n* = 4), GRK2 (*n* = 4), arr3 (*n* = 3) or both (*n* = 4), treated either with 10 μM carvedilol or 10 μM ISO (**c**) and corresponding quantification (**d**). **e, f** Representative NanoBiT-based interaction

between β₂AR-SmBiT and LgBiT-arr2/3 after addition of 10 μM carvedilol or 10 μM ISO (**e**) and corresponding concentration-effect quantifications (*n* = 3, **f**). **g** NanoBiT-based complementation assays in HEK293 wt cells transiently transfected with β₂AR-SmBiT and LgBiT-arr2EE or LgBiT-arr3EE (*n* = 4). **h** BRET-based interaction in HEK293 wt cells between pairs of β₂V₂ C-terminally fused to EYFP (β₂V₂-EYFP) and Renilla luciferase fused to arrestin-3 (RLuc-arr3; *n* = 3), as well as NanoLuc®-labelled β₂V₂ (β₂V₂-NanoLuc®) and Halo-tagged arrestin-3 (Halo-arr3; *n* = 9), after addition of ISO or carvedilol. Kinetic traces are shown as mean + SD of one representative experiment, measured at least in duplicate. Summarized data is depicted as mean ± SEM. **d:** two-way ANOVA with Tukey's post hoc test. Scale bar is 20 μm. Source data are provided as a Source Data file.

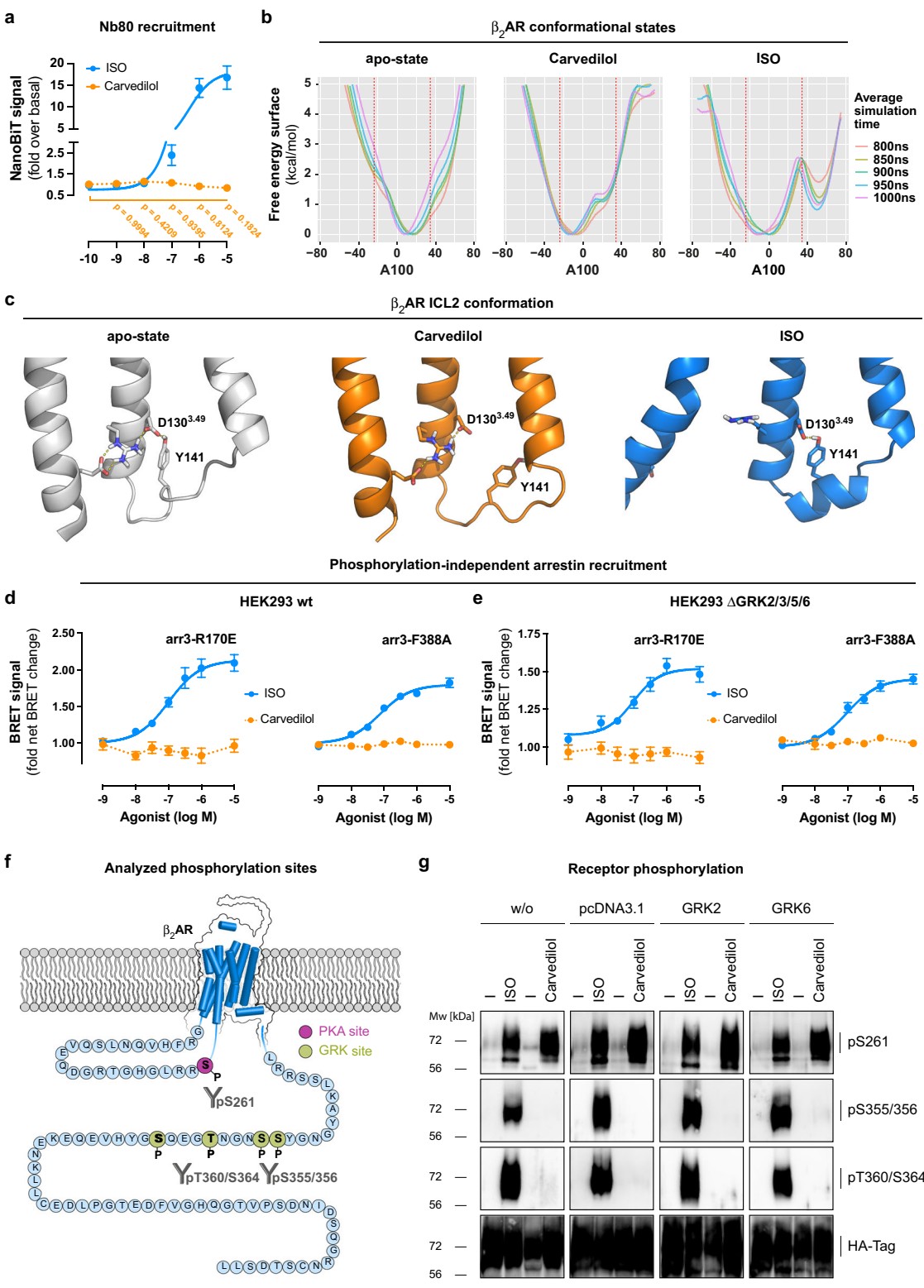

provided by Laura Kilpatrick, Nevin Lambert and Kevin Pfleger, respectively. Nb80-LgBiT corresponds to the $\beta_2$AR active state-stabilizing G protein-mimicking llama-derived single-chain antibody fragment, Nanobody 80 (Nb80) (Q1-S121)[96] N-terminally fused to LgBiT and separated by a 15-residue Ser/Gly linker. The SNAP-tagged $\beta_2$ receptor harbouring the C-terminal tail of the vasopressin $V_2$ receptor (denoted as $\beta_2V_2$) was generated by PCR using pSNAP-$\beta_2$AR and $V_2$R as templates; the C-tail of the $\beta_2$AR was exchanged at position 342 (after

Cys-341 of the $\beta_2$AR) with the last 29 amino acids (Ala-343 to Ser-371) of the $V_2$R. AC5-SmBiT was created by fusing the SmBiT fragment (VTGYRLFEEIL) to the N-terminus of the human full-length adenylyl cyclase isoform 5 (ADCY5) with a 15-amino-acid flexible linker (GGSGGGGSGGSSSGG). The resulting AC5-SmBiT fusion protein was inserted into the pCAGGS expression plasmid. The $G\alpha_s$-LgBiT construct, was previously described[123]. The Halo-tagged arr3 mutants R170E and F388A were generated with the QuikChange site-directed

**Fig. 6 | Ligand-specific β₂AR phosphorylation and molecular dynamics simulations fully explain carvedilol's signalling profile. a** NanoBiT-based enzyme complementation between the Gs-mimetic Nb80-LgBiT and β₂AR-SmBiT upon treatment with carvedilol or ISO. **b** Metadynamics simulations of activation states of the β₂AR. Free energy surfaces (FES), revealing energetically favourable conformations at their respective minima, of unliganded apo-, carvedilol- and ISO-bound β₂AR plotted against the A100 activation index, a measure of receptor activation. The vertical dashed lines indicate the value of A100 for active (at A100 of 34.4) and inactive (at A100 of −24.1) conformations taken from crystal structures, respectively, and are provided for comparison. **c** Representative frames of selected ICL2 clusters from minima of apo- (second cluster, 20.7% of the frames), carvedilol- (second cluster, 32.6% of the frames) and ISO-bound (first cluster, 93% of the frames). The first clusters of apo simulations consist of 63.7% of the frames, and 59.7% for carvedilol-bound, respectively. Conformations in both clusters contain mostly alpha-helical ICL2. Thus, whereas in almost all ISO-bound conformations ICL2 is alpha-helical, apo- and carvedilol-bound simulations contain a significant fraction of conformations in which ICL2 is unstructured. **d, e** NanoBRET-based quantification of proximity between β₂AR-NanoLuc® and phosphorylation-independent Halo-arr3-R170E or Halo-arr3-F388A in HEK293 wt cells (**d**) and the quadruple GRK2/3/5/6 KO HEK293 cell line (aka ΔQ-GRK in ref. 88) (**e**) after treatment with carvedilol or ISO. **f** Snake plot spanning the entire sequence of the β₂AR including the intracellular loop 3 (ICL3) and the receptor C-terminus. The PKA phosphorylation site S261 in the ICL3 is highlighted in dark magenta, and the C-terminal GRK phosphorylation sites S355, S356, T360 and S364 are marked in light green. **g** Characterization of β₂AR phosphorylation with phosphosite-specific antibodies against pS261, pS355/356 and pT360/364 in HEK293 wt cells stably expressing 3xHA-tagged β₂ARs after treatment with 10 μM carvedilol or 10 μM ISO in the absence or presence of overexpressed GRK2 or GRK6. Summarized data are shown as means ± SEM (**a**: $n = 5$; **d, e**: $n = 9$). **a**: one-way ANOVA with Dunnett´s multiple comparisons post hoc test. The blot in **g** is representative of at least three independent experiments with similar results. Source data are provided as a Source Data file.

mutagenesis method using arrestin-3[88] as a template. Human arrestin-3 N-terminally tagged with *Renilla reniformis luciferase II* (RLucII) in pcDNA3.1 was a kind gift of Jakob Lerche Hansen, University of Copenhagen. The Epac1-camps sensor was provided by Jesper Mosolff Mathiesen, University of Copenhagen. All other plasmids used in this study have been previously described: MiniGi- and MiniGs-LgBiT[48], LgBiT-arr2, LgBiT-arr3, LgBiT-arr2R393E, R395E (=LgBiT-arr2EE), LgBiT-arr3R393E, R395E (=LgBiT-arr3EE)[124], Gαs-LgBiT which contains the LgBiT fragment following amino acid position 99 in the human Gαs short-isoform[123], and the plasmids coding for GRK2, GRK6, β₂V₂-NanoLuc and Halo-arr3[88].

## Cell culture

wt HEK293 were obtained from ThermoFisher and chosen as the parental line for CRISPR/Cas9 genome editing. HEK293 lines lacking arrestin-2/-3 (a.k.a. β-arrestin-1/2, Δarr2/3 cells) or the six Gα subunits Gαs/olf/q/11/12/13 (denoted as Δsix) were generated and characterized as previously described in detail[50,125]. Parental wt HEK293 and CRISPR/Cas9 genome-edited HEK293 cells were cultured in DMEM supplemented with 10% foetal bovine serum (FBS, PAN biotech), 100 U per ml penicillin and 100 mg per ml streptomycin at 37 °C in a humidified atmosphere of 95% air and 5% $CO_2$.

Stable SNAP-β₂AR expressing HEK293 cell lines were generated in cooperation with the group of Prof. Dr. Hanns Häberlein (Department of Biochemistry, Bonn University, Germany), as previously reported[126]. The SNAP-β₂V₂ cell line was generated by transfecting wt HEK293 cells with SNAP-β₂V₂ using PEI, at a DNA:PEI ratio of 1:3 (5000 ng DNA, $2.5 × 10^6$ cells). 48 h post-transfection, cells were detached and seeded into a 96-well plate at a density of 1 cell per well in culture media supplemented with 0.5 mg per ml G418. The medium was changed every 48 h. Cells that grew to confluency were detached and expanded stepwise into a 75 cm² flask. Only clones producing high levels of receptor expression (analyzed with anti-SNAP-tag antibody, dilution 1:1000) were selected. All stable cell lines were cultivated using a medium containing 0.5 mg per ml G418.

Generation and culture of CRISPR/Cas 9-genome-edited GRK2/3/5/6 quadruple knockout HEK293 cells (a.k.a. ΔQ-GRK HEK293) has been previously detailed[88].

HEK293 wt cells were stably transfected with a plasmid coding for an N-terminally triple-HA-tagged β₂AR (AR0B20TN00, www.cDNA.org) using TurboFect (Thermo Fisher Scientific) and single-cell clones were expanded by limiting dilution using 400 μg/ml G418 sulfate.

Chinese hamster ovary K1 (CHO-K1) cells were obtained from ATCC and cultured in phenol red-free DMEM/F12, supplemented with 10% FBS, penicillin and streptomycin at 37 °C in a humidified atmosphere of 95% air and 5% $CO_2$.

All cell lines were checked for and used free of mycoplasma contamination. For Gα$_{i/o}$ inhibition analyses, cells were incubated with 100 ng per ml PTX for at least 16 h.

## Membrane preparation

HEK293 cells, stably expressing SNAP-β₂AR were grown to confluency in T175 culture flasks, washed once with ice-cold phosphate-buffered saline (PBS) and scraped off the bottom of the flask in ice-cold PBS. After centrifugation at $12,000 × g$ for 7.5 min at 4 °C, the cell pellet was re-suspended in ice-cold HME buffer (20 mM HEPES pH 7.4, 2 mM $MgCl_2$, 1 mM EDTA) and subjected to one freeze/thaw cycle with liquid nitrogen. The thawed samples were then further homogenized by sonication using a Branson Sonifier Cell Disruptor B15 (output control = 3, duty cycle = 30%, pulsed mode) with $10 × 5$ pulses, and membranes were sedimented by centrifugation for 30 min at $12,000 × g$ at 4 °C. The supernatant was discarded, and cell pellets were re-suspended in ice-cold HME buffer. Membrane emulsions were further homogenized by passing the emulsion several times through a syringe with a diameter of approximately 0.6 μm. The total protein concentration of the membrane preparations was determined with a Pierce BCA Protein Assay Kit (Thermo Fisher) according to the manufacturer's protocol. Membrane samples were aliquoted, snap-frozen in liquid nitrogen and stored at −80 °C.

## Saturation binding assay

Binding reactions with increasing concentrations of [³H]-DHA, ranging from 0.01 pM to 10 μM, were carried out in a final volume of 200 μl binding buffer (HBSS supplemented with 20 mM HEPES pH 7.4) containing 5 μg membrane proteins and 500 μg wheat germ agglutinin (WGA)-coated polyvinyltoluene (PVT) scintillation proximity assay (SPA) beads (PerkinElmer). The binding reaction was carried out for 120 min at room temperature under gentle agitation and terminated by centrifugation at $4500 × g$ at room temperature for 10 min. Non-specific binding of [³H]-DHA was measured in the presence of 1 μM ICI. Radioactivity was quantified using single-photon counting on a TopCount NXT microplate scintillation and luminescence counter (Packard). Data were fitted to the equations of linear regression (one site, total and non-specific binding, respectively) using GraphPad Prism software. To obtain specific binding curves, non-specific binding of [³H]-DHA was subtracted from the total binding signal.

## Competition binding assay

Equilibrium binding was carried out in a binding buffer at a final volume of 200 μL containing 5 μg of membrane proteins, 500 μg of WGA-coated PVT SPA beads, 1 nM [³H]-DHA, and ISO or carvedilol at the indicated concentrations. Non-specific binding of 1 nM [³H]-DHA

was measured in the presence of 1 µM ICI. The binding reaction was carried out for 120 min at room temperature under gentle agitation and terminated by 10 min centrifugation at 453 × g at room temperature for 10 min. Radioactivity was quantified using single-photon counting on a TopCount NXT microplate scintillation and luminescence counter.

To obtain the competition binding curves of ISO and carvedilol, the non-specific binding was subtracted from the total binding to obtain specific binding values. Binding curves were normalized to the maximal response of specific binding. To calculate the $K_i$ values, the data were fitted using nonlinear regression using GraphPad Prism. $pK_i$ was defined as $-logK_i$.

### HTRF ERK1/2 phosphorylation assay

Compound-induced changes in phosphorylated ERK1/2 were quantified using the HTRF technology (Cisbio) following the two-plate protocol for adherent cells according to the manufacturer's instructions. Briefly, HEK293 cells stably expressing SNAP-$\beta_2$AR were seeded into poly-D-lysine (PDL)-coated 96-well plates at a density of 75,000 cells per well and allowed to attach overnight. Where indicated, 100 ng per ml PTX was added to the medium. The next day, the medium was changed to starvation medium (culture medium without FCS), and after 4 h at 37 °C compounds were added for the indicated time points. Cells were subsequently lysed by exchanging the medium with lysis buffer and incubating for 30 min on an orbital shaker. Then, the plates were sealed and frozen at −20 °C. The following day, lysates were thawed, transferred into a white 384-well plate and incubated with the antibody mixtures supplied by the manufacturer for detection of either phosphorylated pERK1/2 or total tERK1/2. The microtiter plate was kept in the dark for at least 2 h prior to reading out with a Mithras² equipped with emission filters at 665 and 620 nm (Berthold Technologies). pERK time courses were analyzed using GraphPad Prism, and data are presented as fold increase over basal, relative to the respective buffer values for each individual time point.

### Label-free dynamic mass redistribution (DMR) assay

DMR was recorded as previously described in detail[127]. Briefly, cells were seeded into fibronectin-coated 384-well Epic biosensor plates (Corning) at a density of 18,000 cells/well and cultivated overnight. The next day, cells were washed twice using HBSS supplemented with 20 mM HEPES and allowed to equilibrate for 60 min at the Epic reader (Corning). After equilibration, measurement was initiated, recording 5 min of baseline read before the addition of compounds using the semi-automated handling system Cybio-Selma (Analytik Jena). Cell DMR triggered by the compounds was recorded for at least 60 min at 37 °C. Where indicated, antagonists were added during equilibration. Raw data were processed using the microplate analyzer (Corning) and analyzed using GraphPad Prism. DMR real-time traces are presented as integrated pm shifts over time following ligand exposure.

### HTRF cAMP accumulation assay

cAMP accumulation was measured using the HTRF technology (Cisbio) and a suspension cell-based protocol. In brief, cells were detached, re-suspended in stimulation buffer (HBSS supplemented with 20 mM HEPES and 1 mM IBMX) and seeded into a white 384-well microtiter plate. To keep the HTRF ratios within the assay dynamic range, the cell number was adjusted to accommodate the efficacy of the different agonists (for Isoproterenol, 500 cells per well; for Carvedilol, 7500 cells per well). The cells were allowed to equilibrate in the plate for 20 min at 37 °C before compound addition. After 30 min incubation at 37 °C, the assay was terminated by the sequential addition of d2-labelled cAMP and cryptate-labelled anti-cAMP antibody, then leaving the plate in the dark for at least 1 h. HTRF ratios were measured using Mithras² (Berthold Technologies) at 665 and 620 nm. Subsequently, HTFR ratios were adjusted at 500 cells to match the conditions of both

agonists and converted to cAMP concentrations using a cAMP standard curve previously assessed according to the manufacturer's protocol.

### Real-time cAMP detection

Ligand-mediated dynamic changes of intracellular cAMP levels were monitored using the Epac1-based fluorescence resonance energy transfer (FRET) cAMP biosensor as previously described[128]. SNAP-$\beta_2$AR stable cell lines were transiently transfected with the EPAC1-camps sensor using Lipofectamine 2000 according to the manufacturer's protocol. Twenty-four hours after transfection, the cells were transferred into a poly-L-lysine-coated black 96-well microplate at a density of 50k cells per well and cultured overnight. Cells were incubated at room temperature in 80 µl HBSS (supplemented with 20 mM HEPES, pH 7.4) in the PHERAstar FSX multimode plate reader (BMG Labtech) for 10 min before stimulation with compounds. The plate was read, measuring the fluorescence of mCerulean (480 nm) and mCitrine (530 nm) upon mCerulean excitation (430 nm) with a dual emission fluorescence optical module (FI 430 530 480). Variations in intracellular cAMP abundance were followed as changes in the mCerulean/mCitrine FRET ratio. FRET ratios were plotted against time during the 60 min post-stimulation for each compound concentration. $EC_{50}$ values were determined in GraphPad Prism from the area under the curve (AUC) values of the different concentration-response curves plotted against ligand concentrations. $pEC_{50}$ vales were defined as $-logEC_{50}$.

### Laboratory animals and ethical statement

Neonatal mice (CD1 background) and 8–10-weeks-old pmEpac1-camps transgenic mice[65] were used. All animal experiments were performed according to institutional and governmental guidelines and approved by the national state authority BGV Hamburg (approval No. ORG_1010). Special approval for harvesting neonatal cardiomyocytes was not required. Housing conditions: mice were kept in 12/12 h dark/light cycles with food and water ad libido at room temperature (20–24 °C) and standard humidity (45–60%).

### cAMP dynamics assay in adult murine ventricular myocytes

Adult ventricular cardiomyocytes were isolated from pmEpac1-camps transgenic mice[65] via retrograde perfusion and enzymatic digestion of the hearts in a Langendorff set-up and seeded on laminin-coated round glass cover slides for live imaging on the same day as previously described[129]. Thus, the cover slides with attached ventricular cardiomyocytes were treated with FRET buffer (144 mM NaCl, 5.4 mM KCl, 1 mM MgCl₂, 1 mM CaCl₂, 10 mM HEPES, pH 7.3) in an Attofluor microscopy chamber on a custom-made Nikon Eclipse Ti microscopy system (Nikon Düsseldorf, Germany) equipped with a ×63/1.40 oil-immersion objective[130]. Upon compound addition, the system recorded cAMP signals generated by the donor fluorophore CFP (excitation at 440 nm) every 5 s using a CoolLED single-wavelength light-emitting diode. Emitted light was detected by an ORCA-03G charge-coupled device camera (Hamamatsu, Herrsching am Ammersee, Germany) after having been split into CFP and YFP channels using a DV2 DualView (Photometrics, Surrey, BC, Canada)[130]. Data analysis was performed using ImageJ/Fiji and GraphPad Prism softwares.

### CardioExcyte96-impedance measurements

Neonatal mouse ventricular cardiomyocytes were dissociated from 1–3-day-old CD1 mouse hearts using the Neonatal Heart Dissociation Kit (Miltenyi Biotec) according to the manufacturer's instructions. Cells were seeded on fibronectin-coated CardioExcyte 96-well sensor plates (Nanion Technologies) in Iscove's Modified Dulbecco's Medium (IMDM) containing 20% FCS, 1% non-essential amino acids, 1% penicillin/streptomycin and 0.1% β-

mercaptoethanol, at a density of 50,000 cells per well. One day later, the medium was exchanged with IMDM containing 2% FCS to prevent fibroblast overgrowth. After 5–7 days, beating frequencies were measured by recording impedance signals with the CardioExcyte96 system (Nanion Technologies) for 30 s every 2–5 min. After 10 min of equilibration in 200 μl IMDM and baseline recording either without blockers or in the presence of ICI (10 μM) and/or CGP (10 μM) cells were stimulated by adding 20 μl pre-heated IMDM containing ISO (final concentration: 300 nM) or carvedilol (final concentration: 300 nM). Baseline frequency and effect of ISO was averaged for 10 min using the CardioExcyte96 control software (Nanion); the effect of carvedilol was shorter lasting and thus averaged over 6 min. Statistical analysis of ISO or carvedilol effects was performed with Student's $t$-test and comparison of relative frequency increases in the presence/absence of blockers was performed with one-way ANOVA repeated measurements and Tukey post-test using GraphPad Prism software (Version 8.43).

**Fluorescence microscopy**
SNAP-$\beta_2$AR expressing HEK293 cell lines were seeded into PDL-coated eight-well plates (Ibidi) at a density of 120,000 cells per well. When necessary, cells were transfected 24 h earlier with GRK2, arrestin-3 or a combination of both (5000 ng DNA, $2.5 \times 10^6$ cells) using PEI at a DNA:PEI ratio of 1:3 as previously reported[131]. The next day after seeding, the medium was exchanged to a solution of SNAP-Surface Alexa Flour 649 in serum-free DMEM and incubated for 30 min at 37 °C and 5% $CO_2$. Afterwards, the plates were washed three times with HBSS, and 150 μl HBSS was added to each well. Images were then acquired on a Zeiss AxioObserver Z (Carl Zeiss, Jena, Germany), equipped with ApoTome2.0 using ×63 magnification. Focal height was maintained by utilizing Definite Focus (Carl Zeiss, Jena, Germany). After acquiring a baseline image, cells were stimulated by adding 150 μl of a 2× agonist solution in HBSS. After the indicated times, a new image was acquired. Image processing and line scan analysis were performed using Zen blue Imaging software (Carl Zeiss, Jena, Germany).

**Diffusion-enhanced resonance energy transfer (DERET)**
Analysis of agonist-induced disappearance of SNAP-$\beta_2$ARs from the cell surface was performed by DERET as described earlier[79–81]. Briefly, HEK293 cells stably expressing SNAP-$\beta_2$ARs were detached and seeded into PDL-treated white 96-well plates at a density of 100,000 cells per well and incubated at 37 °C under humidified conditions with 5% $CO_2$ one day prior to the assay. For overexpression experiments, cells were transiently transfected 24 h earlier, as described above. On the day of the assay, the medium was exchanged to 100 nM SNAP-Lumi4-Tb labelling reagent (Cisbio) in assay buffer (HBSS supplemented with 20 mM HEPES) and incubated for 1 h at 37 °C. Excess SNAP-Lumi4-TB labelling medium was subsequently removed, and the cells were washed four times with assay buffer, followed by the addition of 20 μM fluorescein in the assay buffer. Cells were stimulated by the addition of an equivalent volume of ligands diluted in assay buffer. Fluorescence of Lumi4-Tb (donor, 620 nm) and fluorescein (acceptor, 520 nm) was measured simultaneously using the PHERAstar FSX multimode plate reader after excitation at 337 nm and recorded for at least 30 min. During the assay, the heating chamber and all solutions used were pre-heated to 37 °C to minimize temperature fluctuations. Receptor internalization was assessed by calculating the donor/acceptor ratio using MARS data analysis software (BMG labtech), and the data were fitted to a one-phase exponential association/decay model using GraphPad Prism.

**Clathrin-coated pits (CCP) trapping analysis**
CCP trapping of SNAP-$\beta_2$AR was recorded as previously described[55]. Briefly, CHO-K1 cells were seeded onto ultraclean 25 mm round glass coverslips at a density of $3 \times 10^5$ cells per well. On the following day, cells were transfected using Lipofectamine 2000 with SNAP-$\beta_2$AR and N-terminally GFP-tagged clathrin light chain (GFP-CCP) (kindly provided by Emanuele Cocucci and Tom Kirchhausen), following the manufacturer's protocol. Cells were labelled with 1 μM SNAP-Surface Alexa Flour 549 in complete culture medium for 20 min at 37 °C and used 4 h after transfection to obtain low physiological protein expression levels. Cells were washed with a complete culture medium and imaged in HBSS supplemented with 10 mM HEPES after incubation with compounds. Single-molecule microscopy experiments were performed using total internal reflection fluorescence (TIRF) microscopy as previously described[132]. The sample and objective were maintained at 37 °C throughout the experiments. Multicolour single-molecule image sequences were acquired simultaneously at the full frame in frame transfer mode, corresponding to one image every 30 ms. Automated single-particle detection and tracking were performed with the u-track software[133], and the obtained trajectories were further analyzed using custom algorithms in MATLAB environment as previously described[134]. CCP detection was performed by applying a frame-by-frame binary mask to GFP-CCP image sequences. Sub-trajectory analysis of trapped and free portions was performed using the method based on recurrence matrix[132].

**NanoBiT**
The NanoBiT-based enzyme complementation assay was carried out according to previously published protocols[135,136]. Cell cultures ($2.5 \times 10^6$ cells) were transiently transfected 24 h before the experiment with the respective components (5000 ng total DNA amount) using PEI. When necessary, the DNA mixture was supplemented with empty vector to achieve the final amount of DNA.

Mini-Gs recruitment was detected after transfection of 400 ng $\beta_2$AR-SmBiT and 40 ng mini-Gs-LgBiT.

For complementation assays between $G\alpha_s$-LgBiT and AC5-SmBiT, 500 ng $G\alpha_s$-LgBiT and 500 ng AC5-SmBiT were co-transfected with 500 ng $G\beta_1$, 500 ng $G\gamma_2$ and 100 ng Ric8B.

Arrestin recruitment to $\beta_2$AR was measured after transfecting 600 ng of $\beta_2$AR-SmBiT and 300 ng of either arr2-, arr3-, arr2EE- or arr3EE-LgBiT.

Nb80 recruitment was carried out after transfection with 400 ng $\beta_2$AR-smBiT and 40 ng Nb80-lgBiT.

On the day of the experiment, cells were harvested, re-suspended in assay buffer (HBSS supplemented with 20 mM HEPES and 0.01% BSA or OptiMEM for mini-Gs and Nb80 recruitment) and seeded in a 96-well plate (80,000 cells per well). Coelenterazine (Carbosynth) was added to the cells and incubated in the dark for 2 h at room temperature. For $G\alpha_s$:AC5 interactions and for mini-Gs and Nb80 recruitment, cells were instead incubated with NanoGlo LiveCell substrate diluted 200-fold in assay buffer for 15 min at 37 °C. Compound addition was performed after measuring six time points of baseline read, and the signal was followed for at least 15 min using PHERAstar FSX (BMG Labtech) or Mithras LB940 (Berthold Technologies) readers. Protein-protein interaction (PPI) was calculated as a fold increase of luminescence over baseline and fitted to a one-phase exponential association model using GraphPad Prism.

**Bioluminescence resonance energy transfer (BRET)**
For recruitment examination of mini $G\alpha_s$ or arr3 to $\beta_2$AR, wt HEK293 cells were seeded onto a six-well plate at a density of $7 \times 10^5$ cells per well. On the following day, cells were transfected using Lipofectamine 2000 with $\beta_2$AR-NanoLuc alongside N-terminally Venus-tagged Mini-$G\alpha_s$ or N-terminally Venus-tagged arr3, following the manufacturer's protocol. Twenty-four hours post-transfection, cells were detached and re-suspended in FluoroBrite phenol red-free complete media containing 5% FBS and 2 mM of L-glutamine. Cells were seeded onto a

PDL-coated 96-well white LumiNunc microplates plate at a density of $1 \times 10^5$ cells per well. Cells were left overnight to attach. Forty-eight hours post-transfection, cell media was replaced with HBSS supplemented with 10 mM HEPES and Nano-Glo in 1:500 dilution. Basal BRET measurements were recorded over an 8 min period using the PHERAstar microplate reader with BMG BRET1 filters: Donor wavelength: 475–30 nm and acceptor wavelength: 535–30 nm at 37 °C. Cells were then stimulated with 10 µM carvedilol, isoproterenol or vehicle (HBSS). The resulting ratiometric BRET signal between the interacting fluorophore and lumiphore was normalized by subtracting the background ratio (535–30 nm emission over 475–30 nm) of the vehicle-treated wells with the matched isoproterenol-treated wells producing a signal defined as the "ligand-induced BRET ratio".

For recruitment analysis of arr3 to $\beta_2V_2$, wt HEK293 cells ($2.5 \times 10^6$ cells) were transiently transfected 24 h before the experiment with 500 ng RLuc-arr3 and 500 ng Flag-$\beta_2V_2$-YFP (5000 ng total DNA amount) using PEI. The next day, cells were detached and seeded at a density of 100,000 cells per well in white 96-well plates. Coelenterazine was added at a ratio of 1:5000 and incubated for 15 min before compound addition. RLuc and YFP emissions were measured simultaneously on the PHERAstar FSX multimode plate reader and recorded for at least 30 min. PPI was assessed by calculating the acceptor/donor ratio using MARS data analysis software (BMG labtech), and the data were plotted using GraphPad Prism.

Recruitment evaluation of Halo-arr3 to $\beta_2V_2$-NanoLuc was carried out in transiently transfected wt HEK293 cells, whereas recruitment of Halo-arrestin-3-R170E or Halo-arrestin-3-F388A to $\beta_2AR$-NanoLuc was performed in the CRISPR/Cas 9-derived GRK2/3/5/6 quadruple knockout HEK293 cell line, a.k.a. ΔQ-GRK HEK293[88] and in HEK293 wt cells transfected with empty lentiCRISPR v2 plasmid used herein as CRISPR control line. In 21 cm² dishes, $1.6 \times 10^6$ GRK2/3/5/6 KO cells or $1.2 \times 10^6$ wt cells were seeded and transfected the next day with 500 ng of either $\beta_2AR$-NanoLuc or $\beta_2V_2$-NanoLuc, 1000 ng of Halo-arrestin-3 constructs, or empty vector to adjust the total transfected DNA to 2,5 mg using Effectene (Quiagen) according to the manufacturer's instructions. After 24 h, 40,000 cells per well were seeded into PDL-coated 96-well plates in the presence of HaloTag® 618 ligand at a ratio of 1:2000. For each transfection, a mock labelling condition lacking the Halo-ligand was seeded. The next day, the cells were washed twice with measuring buffer (140 mM NaCl, 10 mM HEPES, 5.4 mM KCl, 2 mM CaCl₂, 1 mM MgCl₂; pH 7.3). Subsequently, a measuring buffer containing the NanoLuc-substrate furimazine in a ratio of 1:35,000 was added. A Synergy Neo2 plate reader (Biotek) with a custom-made filter (excitation bandwidth 541–550 nm, emission 560–595 nm, fluorescence filter 620/15 nm) was used to measure BRET. The baseline was measured for 3 min, and after the addition of the indicated ligands, the measurements were continued for 5 min. PPI was evaluated as previously reported[88], and the data were plotted using GraphPad Prism.

## β₂AR phosphorylation assay

HEK293 wt cells stably expressing HA-tagged $\beta_2AR$ were transiently transfected with vector, GRK2 or GRK6, seeded onto poly-l-lysine-coated 60 mm dishes and grown to 80% confluence. On the day of the assay, cells were treated with compounds for 10 min and then lysed in detergent buffer [50 mM tris-HCl (pH 7.4), 150 mM NaCl, 5 mM EDTA, 10 mM NaF, 10 mM disodium pyrophosphate, 1% (v/v) Nonidet P-40, 0.5% (w/v) sodium deoxycholate and 0.1% (w/v) SDS in the presence of protease and phosphatase inhibitors (Complete mini and PhosSTOP, Roche Diagnostics). HA-beads (Thermo Scientific) were added to the lysates and gently incubated at 4 °C on a turning wheel for 2 h. Proteins were eluted from the beads using SDS sample buffer (125 mM Tris pH 6.8, 4% SDS, 10% glycerol, 167 mM DTT) for 30 min at 50 °C. Samples were separated by 7% SDS-polyacrylamide gel electrophoresis, transferred to nitrocellulose by electroblotting, and $\beta_2AR$ phosphorylation was detected using the phospho-specific antibodies pSer261-$\beta_2AR$ (dilution 1:200), pS355/pS356-$\beta_2AR$ and pT360/pS364-$\beta_2AR$ (dilution 1:200). Total amount of $\beta_2AR$ was detected with the anti-HA-antibody (dilution 1:1000).

## Metadynamics simulations

Active (PDB ID: 3SN6) and inactive (PDB ID: 2RH1) states of $\beta_2AR$ 3D structures were obtained from the GPCRdb[137]. All intracellular proteins were removed, and three different systems were prepared for both active and inactive states, namely *apo*, carvedilol- and ISO-bound. The ligands were manually docked with MOE2019 (Molecular Operating Environment, Chemical Computing Group) and checked for congruence of binding mode with the corresponding $\beta_1AR$ and $\beta_2AR$ structures. The structures were then embedded in a homogenous POPC membrane and solvated with TIP3P model water molecules and 0.15 nM NaCl, using CHARMM-GUI[138] with CHARMM force fields. Three replicates of each system (active/inactive × *apo*/ carvedilol-/ISO-bound) were then equilibrated in NAMD2[139] in six steps. Fewer restraints were applied at each step, as provided by CHARMM-GUI. Afterwards, 100 ns of unbiased molecular dynamics (MD) simulations were performed with ACEMD[140], with a time step of 4 fs, to further equilibrate each system. Hydrogen mass was repartitioned to 4.0 au and constrained during simulations to enable a time step of 4 fs[140,141]. This was followed by well-tempered multiple walker metadynamics simulations performed in ACEMD, with the PLUMED plugin and using the A100 activation index[98] as a collective variable. A Gaussian potential was deposited every 1 ps with a starting height of 1 kJ/mol, a sigma of 1 A100 units, and a bias factor of 10 along a grid of −300 to 300 A100 units, at a spacing of 0.2 A100 units. Multiple walkers consisting of six walkers (3 active and 3 inactive) were used for each ligand-bound state, reading each Gaussian file every 0.4 ps of simulation. Each replicate was run for 1 ms, and the free energy surfaces (FES) were visually inspected at 50 ns × 6 walker intervals (total simulation time of 300 ns per interval, Fig. 6b). The simulations were extended up to 1.5 ms per walker in case we observed visually substantial differences between the intervals, such as changes in shape or difference between minima of the FES. The FES were then constructed from the Gaussian files using PLUMED. Frames from each minimum (±5 A100 units) were extracted and ICL2 backbone atoms were clustered using CPPTRAJ[142]. Representative structures from the minima of each system and JSON files for generating flare plots of hydrogen bonds (occurring in >50% of frames in a minimum) and van der Waals interactions (>80% of frames in a minimum) can be found in the Supplementary Methods for metadynamics simulations and at https://doi.org/10.5281/zenodo.7050831.

## Data and statistical analysis

All data collection has been performed on commercially available software as provided by the manufacturer of the respective device. In particular, MicoWin 2000 (Berthold Technologies GmbH & Co KG, Version: 5.22), Epic Autoalign/Imager Lab View 2009 (Perkin Elmer, Version: 9.0.1f2), PHERAstar FSX reader control (BMG Labtech, Version: 5.41), Zeiss Zen blue edition (Carl Zeiss GmbH, Version: 3.3.89.00000)

Data analysis has been performed on commercially available software. In particular, MARS (BMG Labtech, 3.32), Zeiss Zen blue edition (Carl Zeiss GmbH, Version: 3.3.89.00000). Data and statistical analyses were performed using GraphPad Prism version: 9.1.2, whereas for CardioExcyte96-impedance measurements GraphPad Prism version 8.43 was used. cAMP dynamics assay in adult murine ventricular myocytes was processed in ImageJ/Fiji (https://imagej.net/Fiji).

Concentration-response curves were fitted to a four-parameter equation: Y = Bottom + (Top-Bottom)/(1 + 10^((LogEC50-X)*HillSlope)), where Y is the response, Top and Bottom represent the plateaus of the concentration-response curve in the same units as Y, X is the molar concentration of agonist depicted in log units, $EC_{50}$ is the molar concentration of agonist in log units required to elicit a half-maximal response, and the HillSlope represents a unitless factor. The best-fit value for the HillSlope was challenged with a HillSlope of unity (1) to select the simpler model based on a P-value of 0.05 using an extra sum-of-squares F-test.

Summarized data are presented as the mean ± SEM of 2–4 technical replicates and at least three biological replicates, with the number of independent experiments stated in the figure legend. The data were analyzed by analysis of variance (ANOVA) tests or two-tailed Student's t-test. See figure legends and text for specific statistical analyses used.

## Reporting summary

Further information on research design is available in the Nature Portfolio Reporting Summary linked to this article.

## Data availability

The data that support this study are available from the corresponding author upon request. β2AR 3D structures data used in this study were obtained from the G protein-coupled receptor database (https://gpcrdb.org/structure/3SN6 and https://gpcrdb.org/structure/2RH1). Metadynamics simulations data have been deposited in the open repository Zenodo (https://doi.org/10.5281/zenodo.7050831). All data generated and analyzed during this study are included in this published article and the Supplementary Information. Source data are provided with this paper.

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

## Acknowledgements

We thank Ulrike Rick and Kimberly Harisch for excellent assistance. This study was supported by the Deutsche Forschungsgemeinschaft (DFG, German Research Foundation) 290847012/FOR2372 to E.K. and Heisenberg professorships KO4095/4-1 and KO4095/5-1 to P.K. T.B. was a member of the DFG-funded Research Training Group RTG1873 (214362475/GRK1873/2). A.C. and M.S. were supported by the Luxembourg National Research Fund (INTER/FWO "Nanokine" grant 15/10358798, INTER/FNRS grants 20/15084569, and PoC "Megakine" 19/14209621), F.R.S-FNRS-Télévie (7.8504.20). D.C. was supported by a Wellcome Trust Senior Research Fellowship (212313/Z/18/Z). This article is based upon work from COST Action ERNEST (CA18133), supported by COST (European Cooperation in Science and Technology, www.cost.eu).

## Author contributions

E.K. and J.G. take responsibility for the data integrity and accuracy of data analysis. E.K. and T.B. conceived and designed the study. T.B., M.Z., J.Z., S.B., N.M., E.S.F.M., J.D., E.M.T., D.M., M.S., Ja.Gr., Z.K., Y.L., S.O'B., N.P. and N.D. performed experiments and analyzed the data. V.J.Y.L. and S.M. performed molecular dynamic simulations/modelling and assembled the data. A.I., V.N., D.C., A.C., P.S., S.S., C.H., P.K., M.W., K.S., J.G. and E.K. coordinated, supervised and oversaw the study. T.B. drafted the initial version of the manuscript with support from J.G.; E.K. wrote the manuscript with input from all authors.

## Funding

## Competing interests

S.S. is the founder and scientific advisor of 7TM Antibodies GmbH, Jena, Germany. The remaining authors declare no competing interests.
