## [Peer Review File · Nature Communications]

How Carvedilol activates β 2-adrenoceptorsREVIEWER COMMENTS

Reviewer #1 (Remarks to the Author):

Beta-blockers have been of interest to the molecular pharmacology field for some years since a prominent paper suggested that a potential explanation for the enhanced clinical profile of some blockers vs others could be explained by biased agonism, ie a preference for certain beta1 antagonists to retain beta2 agonist activity at the arrestin pathway. In many ways, this has become dogma in the field.

This manuscript cannot be faulted scientifically and uses an exhaustive set of complementary and confirmatory experimental approaches to ensure that the conclusions drawn are supported by more than one method.

There are two key findings from this paper:

1. The authors definitively demonstrate that the original claims that carvedilol is an arrestin-biased ligand at beta2 adrenergic receptors is not correct and that carvedilol instead has very low efficacy via Gs. The authors are to be commended for acknowledging that the tools to make such a definitive conclusion were not available to the previous authors who reported the original bias study (although some of the other experiments were not reproduced herein). Furthermore, they show that carvedilol is not able to cause a conformational change sufficient to engage arrestins at all (lovely phosphorylation experiments!). This discovery and correction of existing dogma has the potential to be very important for the field.
2. The authors demonstrate that carvedilol is able to activate the beta2-AR in cardiomyocytes while simultaneously inhibiting beta1-ARs. They further expand that this is likely supportive of an older hypothesis that superior beta-blockers contain intrinsic sympathomimetic activity.

My only reservation about the paper is whether these very marginal results, which need to be unmasked by the use of inhibitors of the beta-ARs, are genuinely reflective of the biological situation. I appreciate that every effort has been made to test the hypotheses in biologically relevant cells, and I don't actually have a solution to make it more compelling, I just am having trouble imagining how likely this is to be occurring in vivo.

Minor comments:

- please use IUPHAR nomenclature throughout and be consistent
- for the kinetic traces shown that are representative of a single experiment, it would be more appropriate to represent the data as mean +/- SD, not SEM as indicated
- please put the biosensor label on Figure 3A to aid with interpretation of the figure (as used in Figure 2)

Reviewer #2 (Remarks to the Author):

This manuscript from Benkel and Kostenis demonstrates that carvedilol action at the beta2-adrenoceptor (b2AR) is mediated by Gs stimulation and is independent of arrestins. The team use CRISPR approaches to show the necessity of Gs for carvedilol-induced stimulation of cell activation, ERK phosphorylation and cAMP accumulation. Using BRET (both classical and complementation approaches) and FRET biosensors, they show that carvedilol increases cAMP associated with increased proximity between b2AR and Gs, and Gs and adenylyl cyclase. In cardiomyocytes, carvedilol stimulation of cAMP and increased beating frequency is dependent on activation of the b2AR, and not the b1AR. The team clearly demonstrate that carvedilol induces a receptor conformation that is incapable of recruiting arrestins and incompatible with internalisation (using microscopy, diffusion-enhanced RET, BRET, molecular dynamics and phosphosite-specific antibodies).

This is a very impressive piece of work, and a real tour de force in the use of distinct but complementary experimental approaches to delineate a mechanism of drug action. The figures are clear and well-organised, the methodology is robust and clearly described. This is an important study for the GPCR field and for the therapeutic use of carvedilol.

Below are some comments for the authors to address.

Major comments

1. While the data very strongly support the conclusion that carvedilol activation of the β_2 AR is not compatible with arrestin recruitment and receptor internalisation, it is also very important for the authors to qualify this in the discussion. Typically, arrestin recruitment and receptor internalisation assays show a right-shift in ligand potency compared to assays of G protein activation (e.g. Fig 4D and 5F-H compared to 2A-C). Given that carvedilol is a partial agonist for Gs/cAMP signalling via the β_2 AR, it remains possible that the ligand is a very weak partial agonist for arrestin/internalisation and current assays are just not sensitive enough to show this. The authors also need to make a clear statement that (by necessity) all assays to quantify carvedilol-induced G protein vs arrestin signalling were in over-expression systems, and that results need to be confirmed in endogenous primary cells in the future (when hopefully such techniques may become available).

2. Signalling assays in CRISPR knockout lines (Figure 1): it is interesting that exogenous expression of Gs doesn't completely rescue signalling in the $\Delta 6$ +PTX cells. While it is challenging to get enough Gs expression to completely restore this, there is also an interesting change in the shape of the whole cell activation curve (panel Hi vs Ei). Moreover, the magnitude of the carvedilol-induced cAMP response (a completely Gs-dependent signal) is completely restored (panel L vs I) whereas there seems to be some loss of the maximal ERK phosphorylation signal (panel D vs A; β_2 AR induced ERK can occur via both Gs and Gi/o). Have you done these experiments in the $\Delta 6$ cells in the absence of PTX? As these cells would still have Gi/o proteins, this would reveal any potential role of Gi/o in activating ERK phosphorylation.

Please either include this data as supplementary or discuss this possibility in the manuscript (or do statistical analysis to show no difference in the magnitude of the phospho-ERK signal between the cells).

3. Figure 2: Carvedilol seems to induce a much slower onset of Gs recruitment compared to isoprenaline. Can this difference in apparent rate be quantified?

Similarly, isoprenaline stimulation results in a slow recruitment of arrestins and slow receptor internalisation (on the scale of minutes rather than seconds). Given there is an apparent rate difference between the two ligands at the level of G protein recruitment, it is possible that carvedilol takes much longer than isoprenaline to recruit any arrestins and internalise the receptor. Fig S6 shows no recruitment of arrestins in response to carvedilol up to 60 min post stimulation, which nicely rebuts this possibility. To further strengthen the argument, do you have any DERET data or imaging showing no receptor internalisation in response to carvedilol at a similar timepoint (i.e. beyond the 20 min shown Fig 5)?

4. The Nb80 experiments to demonstrate distinct ligand-induced β_2 AR conformations are very elegant. Have the authors tested any other nanobodies (e.g. the higher affinity active state nb6B9 or inactive

state nb60) or other mini G proteins such as miniGi? Similar data from such experiments would further strengthen this point.

Minor comments

1. Line 174, pg 5: “Indeed, cAMP abundance was enhanced by the action of carvedilol at the β 2AR, and this effect was nullified only in G protein-deficient cells yet recovered with $G_{\alpha s}$ re-expression.” Please correct this statement to “partially recovered”

2. Line 178, pg 5: “Arrestins, on the contrary, were dispensable for initiation of all downstream signals, even those transmitted to pERK, the hallmark feature of arrestin-biased signaling.” Please add references to this statement.

3. Split axes for time courses and concentration-response curves (e.g. Fig 2A, 2C) – please consider changing the range and/or proportion of the split axes. At the moment while the carvedilol data is nicely displayed, the isoprenaline curves are difficult to interpret (in terms of Hill Slope and kinetic profile). In addition, and to fairly represent all data, please add split axis to the graphs that show no effect of carvedilol on arrestin recruitment (e.g. Fig. 5, S6).

4. Figure 3A bar graphs for cAMP response – please include statistical analysis for differences between the antagonist treatment groups for carvedilol. Please include concentrations of antagonists used in these experiments in the figure legends (important for interpretation of receptor sub-type selectivity).

5. Figure S4 – the legend states that the data is from at least 4 technical replicates. Is this correct, and if so, does this mean 4 individual cells from a single preparation? Please make this clear in the legend.

6. Pg 13, line 327: “Of the two possible mechanisms (G proteins vs arrestins) by which carvedilol could drive its activating signals...”. Please rephrase this – while low efficacy G protein signalling is indeed the most likely, it is also possible that G_s vs G_i/o protein bias could play a role, or recruitment of other unidentified effectors, or even carvedilol inducing a unique location of the receptor at the plasma membrane.

7. [related to major point 1] Pg 13, line 348: “Carvedilol induced internalization of at most a tiny fraction of plasma membrane-localized β 2AR over time...” This appears to contradict the last sentence of the next paragraph: “...carvedilol produces a β 2AR conformation, that is signaling competent, yet unable to undergo detectable internalization....” Was the small amount of internalisation referred to in Fig 4D

statistically significant? Is the assay sensitive enough to detect any internalisation in response to carvedilol? Please rephrase and/or comment on this.

8. First paragraph on pg 17, subheading: “Carvedilol-bound b2AR is a poor target for arrestin binding”. The first paragraph here could be moved to the end of the previous section. This is a very nice control experiment to show that there really is no internalisation of the b2AR in response to carvedilol. As the representative DERET trace in Fig 4D shows a small (but not significant) upwards deflection in response to carvedilol, this set of experiments would nicely reinforce the previous conclusions.

9. Pg 17, line 428: “...experimental evidence supporting arrestin recruitment by carvedilol-activated β 2AR was collected with a β 2AR chimera, wherein the C-tail has been swapped with that of the vasopressin V2 receptor (β 2V2)...”

This modification would (obviously) make a big difference in the capacity of a receptor to recruit intracellular binding partners. It could be worth repeating this point when you first mention the discrepancies between your findings (and two other papers), and those of ref 27 (pg 13).

10. Fig 5B: please add “GRK2 +arr3” to the heading of this panel so the reader can easily determine which images the line scan refers to. Please define the timescale on which AUC was calculated for each assay.

11. Is the apparent reduction in DERET in response to carvedilol in Fig S7C significant? If so, please comment in the results, as some antagonists and partial agonists can increase receptor numbers at the plasma membrane.

12. Lack of Nb80 recruitment in response to carvedilol is consistent with previous reports that Nb80 does not shift the affinity of carvedilol for the b2AR (Staus DP et al. 2016 Nature). Please reference this publication.

13. Line 504, pg 21: “Carvedilol, in contrast, does not induce the same active state but, another conformation with molecular features compatible with partial Gs protein activation...” Stabilise may be a more accurate word here, rather than induce.

14. Discussion regarding propranolol – while propranolol does not increase detectable levels of cAMP, it does have some partial agonist activity in reporter gene CRE-SPAP assays (Baker JG et al. 2003 Mol Pharmacol).

Reviewer #3 (Remarks to the Author):

This manuscript describes studies to investigate the functional activity of carvedilol on beta-adrenergic receptors. There is controversy in this area due to differences in the results and interpretations of multiple groups. This is important because carvedilol has reported clinical efficacy superior to other "beta blocker" drugs in some patients suffering from heart failure, and the pharmacological basis for this has been unclear for some time. The present study claims to clarify the pharmacological activity of carvedilol on the beta-2 adrenergic receptor, in particular, through experiments in transfected HEK293 cells and primary embryonic cardiomyocyte cultures.

The manuscript reflects the efforts of an outstanding collaborative team. It contains a large amount of data that appear to be of generally very high quality. The study uses a number of well established methods in an effort to obtain a comprehensive profile of signaling activity. There is also new methodological innovation in the NanoBit and NanoBRET measures of receptor-induced binding of Gs to adenylyl cyclase in intact cells. Overall I am highly enthusiastic about this study. However, I have a few comments that I think might help to improve the paper for the general reader.

1.) I suggest that the interpretation of data could be stated more simply. I believe the conclusion is that carvedilol is a beta-2 conventional partial agonist rather than a beta-arrestin biased agonist? If so, this was hard for me to discern on the first read or two, and it might be useful to make it easier for the reader to "get it".

2.) The main data supporting beta-2 partial agonism is in Fig 2B. I'm guessing the Epac experiment is done under conditions that saturate the sensor because IBMX does not increase the ISO response. Can the authors comment on how efficacious is carvedilol compared to other clinically relevant partial agonists such as salmeterol?

3.) I agree that the fact that CGP20712A unmasks the partial agonist effect in cardiomyocytes supports the hypothesis that this reflects carvedilol having beta-1 inverse agonism. It would be useful to cite here consistency with Galandrin and Bouvier, who reported this conclusion from study of HEK293 cells (Mol Pharm 2006). I don't think this is presently cited.

Reviewer #4 (Remarks to the Author):

Benkel et al. report a very elegant study into the activation of B2 adrenergic receptors by carvedilol. Clearly a lot of work went into the design and performance of a plethora of experiments, that in my opinion provide clear evidence for claims made in the study. I do have a few minor comments regarding the computational work:

The authors need to describe in more detail how protonation states of residues have been treated.

The authors report timesteps of 4 fs, has hydrogen mass repartitioning been performed? This should be made clear in the methods, without any treatment of bonded interactions the simulation would be unstable at the chosen timestep.

The authors mention that simulations have been run until convergence, could they provide more detail about the criteria for convergence?

It would be very helpful if the authors can provide starting structures, input files, and ideally some representative structures of the trajectory, particular since ACEMD is not open source software which makes it hard to reproduce the results.

It would be interesting if the authors could provide some more details about the simulations for instance in supplementary information, for instance overall stability of the system, and a network analysis of key contacts would be very insightful (e.g. as performed in:
<https://www.nature.com/articles/s41592-020-0884-y>)

REVIEWER COMMENTS reproduced verbatim, author comments in blue

Reviewer #1 (Remarks to the Author):

Beta-blockers have been of interest to the molecular pharmacology field for some years since a prominent paper suggested that a potential explanation for the enhanced clinical profile of some blockers vs others could be explained by biased agonism, ie a preference for certain beta1 antagonists to retain beta2 agonist activity at the arrestin pathway. In many ways, this has become dogma in the field.

This manuscript cannot be faulted scientifically and uses an exhaustive set of complementary and confirmatory experimental approaches to ensure that the conclusions drawn are supported by more than one method.

There are two key findings from this paper:

1. The authors definitively demonstrate that the original claims that carvedilol is an arrestin-biased ligand at beta2 adrenergic receptors is not correct and that carvedilol instead has very low efficacy via Gs. The authors are to be commended for acknowledging that the tools to make such a definitive conclusion were not available to the previous authors who reported the original bias study (although some of the other experiments were not reproduced herein). Furthermore, they show that carvedilol is not able to cause a conformational change sufficient to engage arrestins at all (lovely phosphorylation experiments!). This discovery and correction of existing dogma has the potential to be very important for the field.

2. The authors demonstrate that carvedilol is able to activate the beta2-AR in cardiomyocytes while simultaneously inhibiting beta1-ARs. They further expand that this is likely supportive of an older hypothesis that superior beta-blockers contain intrinsic sympathomimetic activity.

My only reservation about the paper is whether these very marginal results, which need to be unmasked by the use of inhibitors of the beta-ARs, are genuinely reflective of the biological situation. I appreciate that every effort has been made to test the hypotheses in biologically relevant cells, and I don't actually have a solution to make it more compelling, I just am having trouble imagining how likely this is to be occurring *in vivo*.

Response authors: We thank this reviewer for the appreciation of our study and their thoughtful comment concerning the relevance of β 2AR activation by carvedilol *in vivo*. As the reviewer states, in primary cardiomyocytes we need to unmask β 2AR activation by concomitant inhibition of β 1AR. However, in failing hearts, for example in ventricular biopsies from patients with heart failure, the patho-biochemical situation is distinct: β 1ARs are downregulated while β 2ARs are not (Milting *et al.*, *J. Mol. Cell. Cardiol.* 2006, PMID:

16765375; Brodde et al., *Pharmacol. Rev.* 1991, PMID: 1677200; Bristow et al., *Mol. Pharmacol.* 1989, PMID: 256462935; Böhm et al., *Cardiovasc. Res.* 1998, PMID: 9876327).

In membranes collected from failing and nonfailing human myocardium (Maack et al., *Br. J. Pharmacol.* 2000, PMID: 10882399) or from a transgenic rat model for left ventricular pressure overload with desensitized beta-adrenergic signal transduction and downregulated β 1ARs (Böhm et al., *Cardiovasc. Res.* 1998, PMID: 9876327), carvedilol shows guanine nucleotide (Gpp(NH)p)-sensitive binding similar to isoprenaline but distinct from metoprolol. Even in healthy rat adult ventricular myocytes carvedilol but not metoprolol revealed “agonist-like” binding characteristics (Flesch et al., *Cardiovasc. Res.* 2001, PMID: 11164847). These features are all indicative of positive intrinsic activity and G protein signaling in the failing myocardium.

In functional experiments using isolated muscle preparations of human left ventricular myocardium from failing hearts, carvedilol increased force of contraction in one and decreased it in six experiments as opposed to metoprolol which decreased force of contraction in all experiments in line with a lack of intrinsic activity (Maack et al., *British J. Pharmacol.* 2000; PMID: 10882399). Moreover, β 1AR and β 2AR distribution in healthy versus failing cardiomyocytes is distinct. In failing cardiomyocytes β 2ARs are redistributed from the T tubules to the cell crest, giving rise to cell wide cAMP propagation patterns similar to those observed for β 1AR which is distributed across the entire cell surface but downregulated in heart failure (Nikolaev et al., *Science* 2010; PMID: 20185685).

Taking all of these independent lines of evidence into account, we infer that intrinsic efficacy of carvedilol via β 2AR is indeed detectable but likely varies from patient to patient depending on (i) the individual numerical changes of β 1AR to β 2AR ratios, (ii) the stage of heart failure, and (iii) the patho-biochemical alterations of the beta-adrenergic receptor system. At present, it is still impossible to predict which patient will benefit from the positive signaling of carvedilol. However, our study makes possible to frame the **molecular basis** underlying the positive signals as **low efficacy G protein** rather than **arrestin-biased mechanism**. As such our discovery and correction of the existing dogma is very important for the field as alluded to under point 1 by this reviewer. We have added additional references and adapted Results (pages 5-6, lines 233-238) and Discussion (page 9, lines 396-399) in support of the notion that the above mentioned *in vitro* findings may likely reflect processes occurring *in vivo*.

Minor comments:

- please use IUPHAR nomenclature throughout and be consistent

Response authors: Thank you for making us aware of using consistent IUPHAR nomenclature. Receptor names were corrected in all instances (see yellow boxed text).

- for the kinetic traces shown that are representative of a single experiment, it would be more appropriate to represent the data as mean +/- SD, not SEM as indicated

Response authors: As requested, kinetic traces, if representative of a single experiment, are now presented as mean + SD (e.g. in Figures 1, 2 or 5).

- please put the biosensor label on Figure 3A to aid with interpretation of the figure (as used in Figure 2)

Response authors: Biosensor label added and Figure inverted to read from left to right.

Reviewer #2 (Remarks to the Author):

This manuscript from Benkel and Kostenis demonstrates that carvedilol action at the beta2-adrenoceptor (b2AR) is mediated by Gs stimulation and is independent of arrestins. The team use CRISPR approaches to show the necessity of Gs for carvedilol-induced stimulation of cell activation, ERK phosphorylation and cAMP accumulation. Using BRET (both classical and complementation approaches) and FRET biosensors, they show that carvedilol increases cAMP associated with increased proximity between b2AR and Gs, and Gs and adenylyl cyclase. In cardiomyocytes, carvedilol stimulation of cAMP and increased beating frequency is dependent on activation of the b2AR, and not the b1AR. The team clearly demonstrate that carvedilol induces a receptor conformation that is incapable of recruiting arrestins and incompatible with internalisation (using microscopy, diffusion-enhanced RET, BRET, molecular dynamics and phosphosite-specific antibodies).

This is a very impressive piece of work, and a real tour de force in the use of distinct but complementary experimental approaches to delineate a mechanism of drug action. The figures are clear and well-organised, the methodology is robust and clearly described. This is an important study for the GPCR field and for the therapeutic use of carvedilol.

Below are some comments for the authors to address.

Response authors: We are very grateful for the positive assessment of our study and the opportunity to further strengthen our findings in writing and by experimentation.

Major comments

1. While the data very strongly support the conclusion that carvedilol activation of the b2AR is not compatible with arrestin recruitment and receptor internalisation, it is also very important for the authors to qualify this in the discussion. Typically, arrestin recruitment and receptor internalisation assays show a right-shift in ligand potency compared to assays of G protein activation (e.g. Fig 4D and 5F-H compared to 2A-C). Given that carvedilol is a partial agonist for Gs/cAMP signalling via the b2AR, it remains possible that the ligand is a very weak partial agonist for arrestin/internalisation and current assays are just not sensitive enough to show this. The authors also need to make a clear statement that (by necessity) all assays to quantify carvedilol-induced G protein vs arrestin signalling were in over-expression

systems, and that results need to be confirmed in endogenous primary cells in the future (when hopefully such techniques may become available).

Response authors: We thank this reviewer for the thoughtful feedback and have included a caveat in the discussion to take into account that most of our analysis was performed with overexpression cells. The caveat reads as follows (Discussion, page 10, lines 443-448): “One caveat that deserves specific mention is that the majority of assays to disentangle G protein versus arrestin-biased signaling were performed in overexpression systems. Hence, extrapolation of our data to primary cells or the *in vivo* situation must be performed with caution and highlight the need to re-investigate the mechanism by which carvedilol exerts its positive signals once comparable techniques become available for primary cells or even whole animals.”

2. Signalling assays in CRISPR knockout lines (Figure 1): it is interesting that exogenous expression of Gs doesn't completely rescue signalling in the delta6+PTX cells. While it is challenging to get enough Gs expression to completely restore this, there is also an interesting change in the shape of the whole cell activation curve (panel Hi vs Ei). Moreover, the magnitude of the carvedilol-induced cAMP response (a completely Gs-dependent signal) is completely restored (panel L vs I) whereas there seems to be some loss of the maximal ERK phosphorylation signal (panel D vs A; b2AR induced ERK can occur via both Gs and Gi/o). Have you done these experiments in the delta6 cells in the absence of PTX? As these cells would still have Gi/o proteins, this would reveal any potential role of Gi/o in activating ERK phosphorylation.

Please either include this data as supplementary or discuss this possibility in the manuscript (or do statistical analysis to show no difference in the magnitude of the phospho-ERK signal between the cells).

Response authors: we have generated four additional biological replicates to better resolve the early phase of the pERK kinetic. These new data clarify the issue of an apparent partial functional rescue upon G α s re-expression, as they clearly show that the effects of carvedilol in relation to those of ISO are slower and partial in nature in all but G protein-depleted (delta6+PTX) cells. We now include statistical analysis to compare the signaling differences of both ligands in each cellular background (**Fig. 1a-d**) and have adapted the manuscript text accordingly (page 4, lines 140-145).

We also provide pERK data in delta6 cells in the absence of PTX (**new Supplementary Fig. 3**). These data reveal no detectable role of Gi/o in ERK phosphorylation for carvedilol, in agreement with Wang *et al.*, *Nat. Commun.* 2017; PMID: 29167435. Moreover, we provide whole cell activation data for ISO and carvedilol in the absence and presence of PTX (**new Supplementary Fig. 4**) again suggesting no appreciable Gi/o activation by carvedilol.

With all due respect to the reviewer comments, we suggest to refrain from statistical analysis of pERK signals across cells because the mere difference of cell passage levels may cause variability of signaling amplitudes that may be unrelated to the presence of defined

transducer proteins such as Gi/o. We feel comfortable with quantitative comparisons within a defined cellular background and qualitative comparisons across different cell lines.

3. Figure 2: Carvedilol seems to induce a much slower onset of Gs recruitment compared to isoprenaline. Can this difference in apparent rate be quantified?

Response authors: As suggested, we have quantified the apparent rate of Gs recruitment and provide this new analysis in **Supplementary Fig. 6** (and here below) showing two entire sets of kinetic traces for ISO and carvedilol as well as a plot of the apparent on-rate over ligand concentration using a non-split and a split axis view (the latter in response to your minor point 3 to fairly represent all data).

Supplementary Figure 6: NanoBiT-based enzyme complementation between β_2 AR-SmBiT and LgBiT-miniGs after treatment with increasing concentrations of either ISO (a) or carvedilol (b).

Similarly, isoprenaline stimulation results in a slow recruitment of arrestins and slow receptor internalisation (on the scale of minutes rather than seconds). Given there is an apparent rate difference between the two ligands at the level of G protein recruitment, it is possible that carvedilol takes much longer than isoprenaline to recruit any arrestins and internalise the receptor. Fig S6 shows no recruitment of arrestins in response to carvedilol up to 60 min post stimulation, which nicely rebuts this possibility. To further strengthen the argument, do you have any DERET data or imaging showing no receptor internalisation in response to carvedilol at a similar timepoint (i.e. beyond the 20 min shown Fig 5)?

Response authors: In analogy to the former Figure S6 (now Supplementary Fig. 13) which shows no arrestin recruitment up to 60 min, we re-analyzed internalization at 20, 40 and 60

min timepoints. The new imaging data shows clearly detectable and prominent internalization for ISO only (**new Supplementary Fig. 12**). Please note that DERET data are also available **beyond the 20 min** timepoint in Fig. 5c. Here, we followed β 2AR surface abundance over **40 min** and found no indication for carvedilol-mediated internalization.

4. The Nb80 experiments to demonstrate distinct ligand-induced β 2AR conformations are very elegant. Have the authors tested any other nanobodies (e.g. the higher affinity active state nb6B9 or inactive state nb60) or other mini G proteins such as miniGi? Similar data from such experiments would further strengthen this point.

Response authors: Thank you for your curiosity. We have performed additional experiments analyzing proximity between β 2AR and miniGi as well as Nb60 in response to both carvedilol and ISO. We find that carvedilol does not detectably promote β 2AR-miniGi interaction (**new Supplementary Fig. 7**) in line with its lack of measurable β 2AR-Gi coupling in other orthogonal assays such as label-free Dynamic Mass Redistribution-based cell shape change (**new Supplementary Fig. 4**) and pERK (**new Supplementary Fig. 3**).

In agreement with *Staus et al. (Nature 2016, PMID: 27409812)* Nb60, reported to stabilize an inactive β 2AR state without altering the affinity of carvedilol binding to β 2AR, is neither actively recruited to nor separated from β 2AR by treatment of cells with carvedilol. Moreover, ISO treatment of cells does not detectably destabilize β 2AR-Nb60 interaction in line with the strong negative allosteric effect of Nb60 on ISO binding to the purified β 2AR (*Staus et al., Nature 2016, PMID: 27409812*). Because our experiments do not provide new mechanistic insight, we respectfully suggest to not include the new data into our manuscript.

Figure: NanoBiT-based enzyme complementation between the Gs-mimetic Nb80-LgBiT or the negative allosteric Nb60-LgBiT and β 2AR-SmBiT upon treatment with carvedilol or ISO.

Minor comments

1. Line 174, pg 5: “Indeed, cAMP abundance was enhanced by the action of carvedilol at the β 2AR, and this effect was nullified only in G protein-deficient cells yet recovered with G α s re-expression.” Please correct this statement to “partially recovered”

Response authors: corrected.

2. Line 178, pg 5: “Arrestins, on the contrary, were dispensable for initiation of all downstream signals, even those transmitted to pERK, the hallmark feature of arrestin-biased signaling.” Please add references to this statement.

Response authors: done, references 25-27,53 are now cited (page 4, lines 158-160).

3. Split axes for time courses and concentration-response curves (e.g. Fig 2A, 2C) – please consider changing the range and/or proportion of the split axes. At the moment while the carvedilol data is nicely displayed, the isoprenaline curves are difficult to interpret (in terms of Hill Slope and kinetic profile). In addition, and to fairly represent all data, please add split axis to the graphs that show no effect of carvedilol on arrestin recruitment (e.g. Fig. 5, S6).

Response authors: We are grateful for this thoughtful, fair and constructive comment. We have revisited all figures of the manuscript and (i) added non-split axes views in all instances where ISO data are difficult to interpret (**new Supplementary Figures 5, 9 and 18**), (ii) added split axes views to the graphs that show no effect of carvedilol (**new Supplementary Figures 13, 14, 15 and 16**) and changed the proportion of the split axes in the **Fig. 2a** and **new Supplementary Fig. 8**.

4. Figure 3A bar graphs for cAMP response – please include statistical analysis for differences between the antagonist treatment groups for carvedilol. Please include concentrations of antagonists used in these experiments in the figure legends (important for interpretation of receptor sub-type selectivity).

Response authors: done, statistical analysis added to **Fig. 3a** and antagonist concentrations are now included in the legend (yellow boxed text).

5. Figure S4 – the legend states that the data is from at least 4 technical replicates. Is this correct, and if so, does this mean 4 individual cells from a single preparation? Please make this clear in the legend.

Response authors: Thank you for spotting this flaw. We have corrected the legend of the previous Figure S4 (**Supplementary Fig. 10** in the revised document) which now reads “Each data point represents an individual well from at least three different animal preparations”.

6. Pg 13, line 327: “Of the two possible mechanisms (G proteins vs arrestins) by which carvedilol could drive its activating signals...”. Please rephrase this – while low efficacy G protein signalling is indeed the most likely, it is also possible that G α s vs G α i/o protein bias could play a role, or recruitment of other unidentified effectors, or even carvedilol inducing a unique location of the receptor at the plasma membrane.

Response authors: Thank you for broadening our perspective to more than two mechanisms but also for the appreciation that low efficacy G protein signaling may be the most likely. As suggested, we rephrased the statement which now reads “Of the possible mechanisms by which carvedilol could drive its activating signals (G proteins *versus* arrestins or even as yet unidentified effectors), low efficacy G protein signaling is clearly the most likely.” (page 6, lines 239-241)

7. [related to major point 1] Pg 13, line 348: “Carvedilol induced internalization of at most a tiny fraction of plasma membrane-localized β 2AR over time...” This appears to contradict the last sentence of the next paragraph: “...carvedilol produces a β 2AR conformation, that is signaling competent, yet unable to undergo detectable internalization....” Was the small amount of internalisation referred to in Fig 4D statistically significant? Is the assay sensitive enough to detect any internalisation in response to carvedilol? Please rephrase and/or comment on this.

Response authors: Thank you for making us aware of this semantical imprecision and the apparent inconsistency. As suggested, we rephrased to “Carvedilol induced no observable internalization...” as its effect is not statistically significant (page 6, lines 262-263). Moreover, related to point 3 of this reviewer (reproduced verbatim: “To further strengthen the argument, do you have any DERET data or imaging showing no receptor internalisation in response to carvedilol at a similar timepoint (i.e. beyond the 20 min shown Fig 5)?”), we also added new imaging data at 20, 40, and 60 min timepoints clearly indicating no appreciable β 2AR internalization induced by carvedilol (**new Supplementary Fig. 12**).

8. First paragraph on pg 17, subheading: “Carvedilol-bound β 2AR is a poor target for arrestin binding”. The first paragraph here could be moved to the end of the previous section. This is a very nice control experiment to show that there really is no internalisation of the β 2AR in response to carvedilol. As the representative DERET trace in Fig 4D shows a small (but not significant) upwards deflection in response to carvedilol, this set of experiments would nicely reinforce the previous conclusions.

Response authors: We very much agree that the first paragraph would fit nicely to the previous section. However, Figure 4 is already quite crowded with data that should be presented in the main body manuscript. Therefore, we respectfully suggest to maintain the current section at its present location.

9. Pg 17, line 428: “...experimental evidence supporting arrestin recruitment by carvedilol-activated β 2AR was collected with a β 2AR chimera, wherein the C-tail has been swapped with that of the vasopressin V2 receptor (β 2V2)...”

This modification would (obviously) make a big difference in the capacity of a receptor to recruit intracellular binding partners. It could be worth repeating this point when you first mention the discrepancies between your findings (and two other papers), and those of ref 27 (pg 13).

Response authors: We thank the reviewer for this suggestion. However, we are afraid that it might be misleading to mention utilization of the β 2V2 chimeric receptor when we first mention the internalization discrepancies between our study and the data reported in ref 27. The **internalization** data in **ref 27** were indeed collected with the **β 2AR wildtype** unlike the **arrestin recruitment assays** where the authors utilized the **β 2V2 chimera**. Therefore, we introduce and explain the β 2V2 chimeric construct, when we describe the discrepancies in the arrestin recruitment assays.

10. Fig 5B: please add “GRK2 +arr3” to the heading of this panel so the reader can easily determine which images the line scan refers to. Please define the timescale on which AUC was calculated for each assay.

Response authors: done, GRK2 + arr3 added to **Fig. 5b** and timescales defined on which AUC was calculated for **Fig. 5g** and **5h**.

11. Is the apparent reduction in DERET in response to carvedilol in Fig S7C significant? If so, please comment in the results, as some antagonists and partial agonists can increase receptor numbers at the plasma membrane.

Response authors: The apparent reduction in DERET response is not significant in the former Figure S7C, **now Supplementary Fig. 17** including statistical analysis.

12. Lack of Nb80 recruitment in response to carvedilol is consistent with previous reports that Nb80 does not shift the affinity of carvedilol for the b2AR (Staus DP et al. 2016 Nature). Please reference this publication.

Response authors: done, reference included as no. 97. We also specifically mention that our data are consistent with the Staus study (more precisely, with the Extended data to Fig. 3), page 8, lines 336-337.

13. Line 504, pg 21: “Carvedilol, in contrast, does not induce the same active state but, another conformation with molecular features compatible with partial Gs protein activation...” Stabilise may be a more accurate word here, rather than induce.

Response authors: rephrased to “stabilize” (page 8, line 345).

14. Discussion regarding propranolol – while propranolol does not increase detectable levels of cAMP, it does have some partial agonist activity in reporter gene CRE-SPAP assays (Baker JG et al. 2003 Mol Pharmacol).

Response authors: thank you for highlighting this publication, which is referenced in our manuscript on page 2, line 73 and again on page 10, line 432. In both instances we refer to the partial agonist action of carvedilol or an alternative activating mechanism.

Reviewer #3 (Remarks to the Author):

This manuscript describes studies to investigate the functional activity of carvedilol on beta-

adrenergic receptors. There is controversy in this area due to differences in the results and interpretations of multiple groups. This is important because carvedilol has reported clinical efficacy superior to other "beta blocker" drugs in some patients suffering from heart failure, and the pharmacological basis for this has been unclear for some time. The present study claims to clarify the pharmacological activity of carvedilol on the beta-2 adrenergic receptor, in particular, through experiments in transfected HEK293 cells and primary embryonic cardiomyocyte cultures.

The manuscript reflects the efforts of an outstanding collaborative team. It contains a large amount of data that appear to be of generally very high quality. The study uses a number of well established methods in an effort to obtain a comprehensive profile of signaling activity. There is also new methodological innovation in the NanoBit and NanoBRET measures of receptor-induced binding of Gs to adenylyl cyclase in intact cells. Overall I am highly enthusiastic about this study. However, I have a few comments that I think might help to improve the paper for the general reader.

Response authors: We are grateful for the enthusiastic feedback and the appreciation of our study as well as the opportunity to further improve our manuscript to appeal to and engage a broader scientific audience.

1.) I suggest that the interpretation of data could be stated more simply. I believe the conclusion is that carvedilol is a beta-2 conventional partial agonist rather than a beta-arrestin biased agonist? If so, this was hard for me to discern on the first read or two, and it might be useful to make it easier for the reader to "get it".

Response authors: As the clear classification of carvedilol as conventional partial β_2 agonist rather than an arrestin-biased agonist is a highly sensitive issue, which required tools and technologies that were not previously accessible to the authors, our results need to be communicated in a considerate and thoughtful manner which sometimes suffers from a lack of clarity. Regardless, we cannot agree more with the conclusion of this reviewer: All detectable carvedilol signaling we observed in our study is entirely G protein-driven. To make it easier for the reader "to get it" we rephrased the text in the below instances:

DISCUSSION

page 9, lines 385-387: This was the goal of the present study, and the uncovered mechanism of how carvedilol exerts its positive signaling function through β_2 AR is our major finding.

Rephrased to "This was the goal of the present study, and the uncovered mechanism that carvedilol exerts its positive signaling function through β_2 AR **via low efficacy G protein activation** is our major finding."

page 9, lines 393-396: Our study resolves this conundrum and clarifies, unambiguously, how carvedilol activates β_2 ARs. We find that the unique signaling activity of carvedilol at β_2 AR arises from low efficacy partial activation of heterotrimeric Gs proteins but does, unexpectedly, not require contribution from arrestins. **Rephrased to** "Our study resolves this

conundrum and clarifies, unambiguously, that the unique signaling activity of carvedilol at β 2AR arises from low efficacy partial activation of heterotrimeric Gs proteins but does, unexpectedly, not require contribution from arrestins.

2.) The main data supporting beta-2 partial agonism is in Fig 2B. I'm guessing the Epac experiment is done under conditions that saturate the sensor because IBMX does not increase the ISO response. Can the authors comment on how efficacious is carvedilol compared to other clinically relevant partial agonists such as salmeterol?

Response authors: B2AR **partial agonism** is supported in **Fig. 2b** but also **very clearly** in Figures **1a,b,d,e,f,h,k** and much less so in Fig.1 i,j. From these experiments we infer that the **efficacy** of carvedilol relative to that of ISO **varies** and that this variation depends on the methodology used to assess receptor signaling, the sensitivity of the detection method as well as the amplification within the signaling pathway. **Efficacy** of carvedilol is **particularly prominent** in **highly amplified assays** such as those depicted in **Fig. 1e-h**. Here signaling is converted to a change in cell morphology yielding greater apparent efficacy as compared with assays that measure cAMP abundance. FRET-based cAMP quantification in **Fig. 2b** was performed in the **subsaturating** sensor range for **carvedilol** but at sensor saturation for ISO. This **relevant technical detail** is now explicitly stated in the legend of Fig. 2 (yellow boxed text).

Moreover, we have characterized salmeterol head-to-head with carvedilol and ISO using the same experimental conditions as those in Fig. 2. Under these conditions **carvedilol efficacy** is **lower** as compared with **salmeterol**, a clinically used partial β 2AR agonist, which again differs from ISO by its kinetically distinct pattern of cAMP formation at subsaturating ligand concentration. As **betablockers with too much ISA** may be problematic in the treatment of heart failure (increased mortality), carvedilol's weak agonism at stimulating cAMP production may be of clinical advantage. In relation to this very point, please also see **point 2 of reviewer 1** (likelihood of carvedilol cAMP signaling *in vivo* and our answer highlighting the manuscript alterations in the Results (pages 5-6, lines 233-238) and Discussion section (page 9, lines 396-399) to address the probability that the *in vitro* signaling profile may indeed reflect processes occurring *in vivo*.

For the sake of consistency with all remaining panels detailing the action of **carvedilol in relation to ISO only**, we respectfully suggest to not include our new salmeterol data in the revised manuscript.

Figure: FRET-based detection of cAMP with the cytosolic Epac1-camps sensor. Identical setup as shown in Fig. 2b. Please note the different y-axis scaling for carvedilol.

3.) I agree that the fact that CGP20712A unmasks the partial agonist effect in cardiomyocytes supports the hypothesis that this reflects carvedilol having beta-1 inverse agonism. It would be useful to cite here consistency with Galandrin and Bouvier, who reported this conclusion from study of HEK293 cells (Mol Pharm 2006). I don't think this is presently cited.

Response authors: Thank you for making us aware of this relevant publication which is now included in the reference list as no. 59. Please note that Galandrin & Bouvier (*Mol. Pharm. 2006; PMID: 16901982*) have shown partial β_1 agonism for carvedilol in both cAMP (Fig. 1) and pERK (Fig. 2) assays as well as partial β_2 activation in ERK phosphorylation assays (Fig. 4). Therefore, we reference this publication in our revised manuscript in support of partial (~30%) carvedilol agonism on the ERK/MAPK pathway to acknowledge this early and, at the time, unexpected observation.

Reviewer #4 (Remarks to the Author):

Benkel et al. report a very elegant study into the activation of B2 adrenergic receptors by carvedilol. Clearly a lot of work went into the design and performance of a plethora of experiments, that in my opinion provide clear evidence for claims made in the study. I do have a few minor comments regarding the computational work:

Response authors: We are very grateful for the positive evaluation of our manuscript and the appreciation of our findings.

1. The authors need to describe in more detail how protonation states of residues have been treated.

Response authors: We can certainly provide this information (which can also be obtained from the pdb files that we have submitted in response to one of the following questions of the reviewer). We also added the below information to the Supplementary Section.

β_2 AR protein treatment and protonation state for MD simulations

Histidine protonation (HSD: proton in the δ -nitrogen of the imidazole ring; HSE: proton on the ϵ -nitrogen; HSP: protons on both nitrogen atoms):

- HSP22
- HSE93
- HSE172
- HSP178
- HSP241
- HSP269
- HSD296

Aspartic/Glutamic acid protonation:

- All charged amino acids are (de)protonated such that the side chain carries a formal charge, unless specify otherwise

- ASP79 protonated (i.e. uncharged) to promote stability of active state
- GLU122 protonated (i.e. uncharged) as surrounded by hydrophobic lipid tails

Post-translational modifications:

- CYS341 palmitoylated

N terminal capped with ACE

C terminal capped with CT3

2. The authors report timesteps of 4 fs, has hydrogen mass repartitioning been performed? This should be made clear in the methods, without any treatment of bonded interactions the simulation would be unstable at the chosen timestep.

Response authors: Thank you for pointing out this omission. Indeed, hydrogen mass was repartitioned to 4.0 au and constrained during simulations to enable a time step of 4 fs. (Feenstra, K Anton et al., *J. Comput. Chem.* 1999, doi:10.1002/(SICI)1096-987X(199906)20:8<786::AID-JCC5>3.0.CO;2-B and Harvey, M J et al., *J. Chem. Theory Comput* 2009, doi:10.1021/ct9000685). This sentence has also been added to the Methods section (page 18, lines 845-847).

3. The authors mention that simulations have been run until convergence, could they provide more detail about the criteria for convergence?

Response authors: Each replicate was run for 1 ms and the FES were visually inspected at 50 ns x 6 walker intervals (total simulation time of 300 ns per interval) (Figure 6b). The simulations were extended up to 1.5 ms per walker in case we observed visually substantial differences between the intervals, such as changes in shape or difference between minima of the FES. While we also calculated RMSD values between the different FES, these were double-checked also by eye, so as to avoid those cases where simple shifts in absolute values of A100 would lead to artificially high values of RMSD despite similarly shaped FES. We have added corresponding text to the manuscript (page 18, lines 853-858).

4. It would be very helpful if the authors can provide starting structures, input files, and ideally some representative structures of the trajectory, particular since ACEMD is not open source software which makes it hard to reproduce the results. It would be interesting if the authors could provide some more details about the simulations for instance in supplementary information, for instance overall stability of the system, and a network analysis of key contacts would be very insightful (e.g. as performed in: <https://www.nature.com/articles/s41592-020-0884-y>)

Response authors: Initial starting structures and input files for ACEMD and PLUMED have been uploaded to Zenodo and are available at <https://doi.org/10.5281/zenodo.7050831>. We

also would like to note that while ACEMD is not open-source, academics can run it for free on the first GPU of each node in a cluster. This allows reproduction of our results without the need to obtain a paid-for license.

Frames were extracted from each minimum, aligned, and clustered based on the root mean square distance of C α atoms of the protein. Five clusters were generated for each minimum and a representative structure for each cluster was uploaded. The majority of the frames in each minimum (>50% of the frames) were in the first cluster, indicating that the protein has a relatively stable state in each of the minima, regardless of the bound ligand. The most “unstable” minimum would be for the apo simulations, where only 51% of the frames are in the first cluster, while ISO-bound β 2AR is the most stable in the second minimum (81% of frames in first cluster). Flare plots of hydrogen bonds (occurring in >50% of frames in a minimum) and van der Waals interactions (>80% of frames in a minimum) can be generated from the JSON files also available at <https://doi.org/10.5281/zenodo.7050831> (we did not generate static images for reasons of resolution and because the key aspect of highlighting different interactions would be lost).

Text to this effect has also been added to the **main manuscript** (page 18/19, lines 860-864) and a more detailed description of results and methods to ensure reproducibility of our findings can be found in Supplementary Information.

REVIEWERS' COMMENTS

Reviewer #1 (Remarks to the Author):

I am happy with the response to my comments and the changes made to the manuscript.

Reviewer #2 (Remarks to the Author):

The authors have very thoroughly addressed all the points raised during review. This is an impressive paper that will have a great impact in the field.

Reviewer #3 (Remarks to the Author):

I liked this study before and think the revised manuscript is significantly improved. I have no additional critiques and strongly recommend publication.

Reviewer #4 (Remarks to the Author):

I would like to thank the authors for their careful considerations and replies made to my points. I believe they have sufficiently answered my concerns and therefore recommend to accept this manuscript for publication.

Reviewer #5 (Remarks to the Author):

Benkel and colleagues present a deep study into the mechanisms of carvedilol's pharmacological action. They take multiple complementary approaches to investigate carvedilol mediated signalling through the

β 2AR. This work changes and clarifies what was previously thought about the mechanism of action of carvedilol. This is a significant study for the whole GPCR field.

Some of the elegant results include: 1. The use of multiple CRISPR knockout cell lines to disentangle effector coupling, 2. Unmasking the effect of carvedilol through the β 2AR in cardiomyocytes, 3. Identifying weak Gs (but not arrestin) mediated signalling events by carvedilol in multiple different setups, and, 4. Detection of distinct agonist-dependent receptor phosphorylation patterns.

All minor flaws in the interpretation and analysis I feel have been corrected or addressed in the authors rebuttal. The authors use different but overlapping molecular, cellular, and computational techniques in their methodology, and thus, demonstrate their claims in a robust manner. The authors have sufficiently addressed any concerns to recommend publication.

REVIEWER COMMENTS reproduced verbatim, author comments in blue

Reviewer #5 (Remarks to the Editor):

Here are just two very minor follow-up points that I thought you may like extra clarification on:

Regarding our major comment no. 1:

I apologise, we probably weren't very clear here in this major comment no.1. There are really two parts to this major comment and the authors only specifically addressed the second part of our comment. In our first part, we are referring to the rightward shift in potency (pEC50) values that is observed when comparing arrestin recruitment to Gs protein pathways. For example, the authors show that isoprenaline's potency is drastically shifted rightward in arrestin recruitment (~500nM) (Fig. 5) compared to the Gs protein endpoints measured (~0.1nM) (Fig. 1 & 2). It could be expected that different agonists (including carvedilol) may display this characteristic because there is less amplification in the arrestin assay. Therefore, because carvedilol's potency in Gs protein endpoints is around 100nM, in the arrestin endpoints the pEC50 could theoretically be anywhere around ~1µM-1mM???. However, the maximum concentration of carvedilol the authors use is only 10µM. If the authors had tested carvedilol at concentrations higher than 10µM, one may start to observe responses in the arrestin assay. We understand there could be solubility issues and off-target effects at these higher carvedilol concentrations, therefore some justification or qualification for this may be appropriate.

Response authors: you were very clear in the major comment no.1. Of the two parts to this major comment, one simply escaped our attention. We apologize for this flaw and are thankful for being reminded again, because this comment – like the many others we received – are key to make our paper even more transparent and fair. Of course, we cannot exclude that carvedilol used at exceedingly high concentrations would indeed recruit arrestins. However, 10 micromolar is really the upper limit for carvedilol and is also the highest applied concentration in numerous studies because of solubility and quenching issues. It is also THE concentration used by others to claim internalization and arrestin recruitment, although only a single cell is shown in ref. no. 27 to support the conclusion in the absence of any additional complementary evidence. Regardless, it is certainly fair to qualify that our assays may not be sensitive enough to detect this, and we do this on page 7 lines 317-320 with the following caveat: “As a caveat to our posit, it remains possible that carvedilol is a very slow and low efficacy agonist also for arrestin recruitment and internalisation, but that current assays may lack sensitivity and/or timescale to detect this low efficacy at the maximum possible concentration.”

Regarding the author response to our minor point no. 3:

Iso's maximum rate of miniGs recruitment = ~0.180 RFU per s (Supp Fig 6).

Carv's max rate of miniGs recruitment = ~0.0012 RFU per s (Supp Fig 6).

Therefore, carvedilol has around a ~150-fold slower activation rate for miniGs than isoprenaline.

The point we were trying to express in this minor point no.3 is that, because carvedilol is a partial agonist, it could be expected to also have a ~150-fold slower activation rate (relative to iso) for arrestin recruitment and internalisation. Therefore, quantifying the rate of iso-induced internalisation and extrapolating the expected rate of carvedilol induced internalisation would allow the authors to determine whether they have detected the arrestin recruitment and internalisation at a long enough time period after stimulation to “catch” a potentially very slow/delayed response.

Response authors: The half- life for Iso-induced internalization is ~360 seconds (=6 min). Extrapolating this value to calculate an expected half-life for carvedilol-induced arrestin recruitment or internalization yields a half-life of 900 min = 16 hours. If the extrapolation was correct and G protein signaling and desensitization/internalization responses were indeed comparably slow, which we cannot assume with certainty, none of our assays had been set up to catch this process. Please bear in mind that the same concentration of carvedilol used in our study (10 micromolar) has been applied in ref 27 to detect beta-arrestin2 translocation within just **2 minutes** and **low efficacy internalization (15%** relative to ISO) within only 30 minutes. Thus, it is very unlikely that we will detect internalization and arrestin recruitment if we just wait long enough. Rather than exposing our readers to these details, we feel that the above blue sentence is well suited to state that our assays may have not been set up to catch potentially very slow and low efficacy effects of carvedilol.